# Enhanced Moisture Delivery into Victoria Land, East Antarctica During the Early Last Interglacial: Implications for West Antarctic Ice Sheet Stability

Yuzhen Yan[1,2], Nicole E. Spaulding[3], Michael L. Bender[1,4], Edward J. Brook[5], John A. Higgins[1], Andrei V. Kurbatov[3], Paul A. Mayewski[3]

[1]Department of Geosciences, Princeton University, Princeton NJ 08544, USA
[2]Department of Earth, Environmental and Planetary Sciences, Rice University, Houston TX 77005, USA
[3]Climate Change Institute, University of Maine, Orono ME 04469, USA
[4]School of Oceanography, Shanghai Jiao Tong University, Shanghai 200240, China
[5]College of Earth, Ocean, and Atmospheric Sciences, Oregon State University, Corvallis OR 97331, USA

*Correspondence to*: Yuzhen Yan (yuzhen.yan@rice.edu)

**Abstract.** The S27 ice core, drilled in the Allan Hills Blue Ice Area of East Antarctica, is located in Southern Victoria Land ~80 km away from the present-day northern edge of the Ross Ice Shelf. Here, we utilize the reconstructed accumulation rate of S27 covering the Last Interglacial (LIG) period between 129 and 116 thousand years before present (ka) to infer moisture transport into the region. The accumulation rate is based on the ice age-gas age differences calculated from the ice chronology, which is constrained by the stable water isotopes of the ice, and an improved gas chronology based on measurements of oxygen isotopes of $O_2$ in the trapped gases. The peak accumulation rate in S27 occurred at 128.2 ka, near the peak LIG warming in Antarctica. Even the most conservative estimate yields an order-of-magnitude increase in the accumulation rate during the LIG maximum, whereas other Antarctic ice cores are typically characterized by a glacial-interglacial difference of a factor of two to three. While part of the increase in S27 accumulation rates must originate from changes in the large-scale atmospheric circulation, additional mechanisms are needed to explain the large changes. We hypothesize that the exceptionally high snow accumulation recorded in S27 reflects open-ocean conditions in the Ross Sea, created by reduced sea ice extent and increased polynya size, and perhaps by a southward retreat of the Ross Ice Shelf relative to its present-day position near the onset of LIG. The proposed ice shelf retreat would also be compatible with a sea-level high stand around 129 ka significantly sourced from West Antarctica. The peak in S27 accumulation rates is transient, suggesting that if the Ross Ice Shelf had indeed retreated during the early LIG, it would have re-advanced by 125 ka.

## 1 Introduction

The West Antarctic Ice Sheet (WAIS) is grounded on bedrock that currently lies below sea level, and is therefore vulnerable to rising temperatures (Mercer, 1968; Hughes, 1973). Yet, the stability of WAIS remains poorly understood and constitutes a major source of uncertainty in projecting future sea-level rises in a warming world (Dutton et al, 2015a; DeConto and

Pollard, 2016). One way to constrain the sensitivity of ice sheets to climate change is to explore their behavior during past warm periods. The Last Interglacial (LIG) between 129 and 116 thousand years before present (ka) is a geologically recent warm interval with average global temperature 0 to 2 °C above the pre-industrial level (Otto-Bliesner et al, 2013). The LIG could therefore shed light on the response of WAIS to future warming. While the WAIS must have contributed to the LIG

sea-level high stand (Dutton et al, 2015a and references therein), quantifying these contributions is challenging and the timing of such WAIS changes (early versus late in LIG) is still debated (e.g. Yau et al, 2016; Rohling et al, 2019; Clark et al, 2020).

As the floating extension of land ice masses, the extent of sea ice and ice shelves can provide important insights into the dynamics of continental ice sheets. For example, as the ocean warms and sea level rises, the loss of ice shelves due to

calving and basal melting may lead to further losses of the continental ice they buttress (Pritchard et al, 2012). The Ross Ice Shelf (RIS) is the largest ice shelf in Antarctica, located between the Marie Byrd Land in West Antarctica and the Victoria Land in East Antarctica (Figure 1). Ice sheet models have suggested that the complete disintegration of the Ross Ice Shelf may have accompanied the collapse of WAIS in both past and future simulations (DeConto and Pollard, 2016; Garbe et al, 2020). However, terrestrial evidence is lacking due to subsequent ice sheet growth, and existing marine records do not have

enough temporal resolution to resolve the extent of RIS during the LIG.

Ice cores provide continuous, well-dated records of local climate information that is, in turn, sensitive to the extent of nearby ice masses. The position of the ice margin and sea ice extent can impact atmospheric circulation, snow deposition, and isotopic signatures in the precipitation captured in ice cores (Morse et al, 1998; Steig et al, 2015; Holloway et al, 2016). In this study, we use a shallow ice core, Site 27 (S27) from the Allan Hills Blue Ice Area (BIA), to explore RIS changes during

the LIG. The Allan Hills BIA in Victoria Land, East Antarctica, is ideally located near the present-day northwest margin of the RIS (Figure 1 and Figure S1). S27 provides a continuous climate record between 115 and 255 ka (Spaulding et al, 2013). The close proximity of Site 27 to the Ross Sea embayment holds the potential to shed light on the behavior of the Ross Ice Shelf during Termination II (the transition from the Penultimate Glacial Maximum to the LIG) and, by extension, on the West Antarctic Ice Sheet.

Here, we present a record of the accumulation rate of Site 27 between 115 and 140 ka, derived from independently constrained ice and gas chronologies. This approach has previously been applied to Taylor Glacier blue ice samples to estimate accumulation rates between 10 and 40 ka (Baggenstos et al, 2018) and between 55 and 84 ka (Menking et al, 2019). We take advantage of the fact that the age of the ice is older than the age of the trapped gases at the same depth. This age difference ($\Delta$age) results from the process of converting snow into ice (firn densification) and reflects the age of the ice when

the firn crosses a threshold density where the gases become isolated in impermeable ice. The evolution of firn density is found to empirically correlate with the ice accumulation rate and surface temperature (Herron and Langway 1980). Subsequent ice thinning and flow do not alter this $\Delta$age.

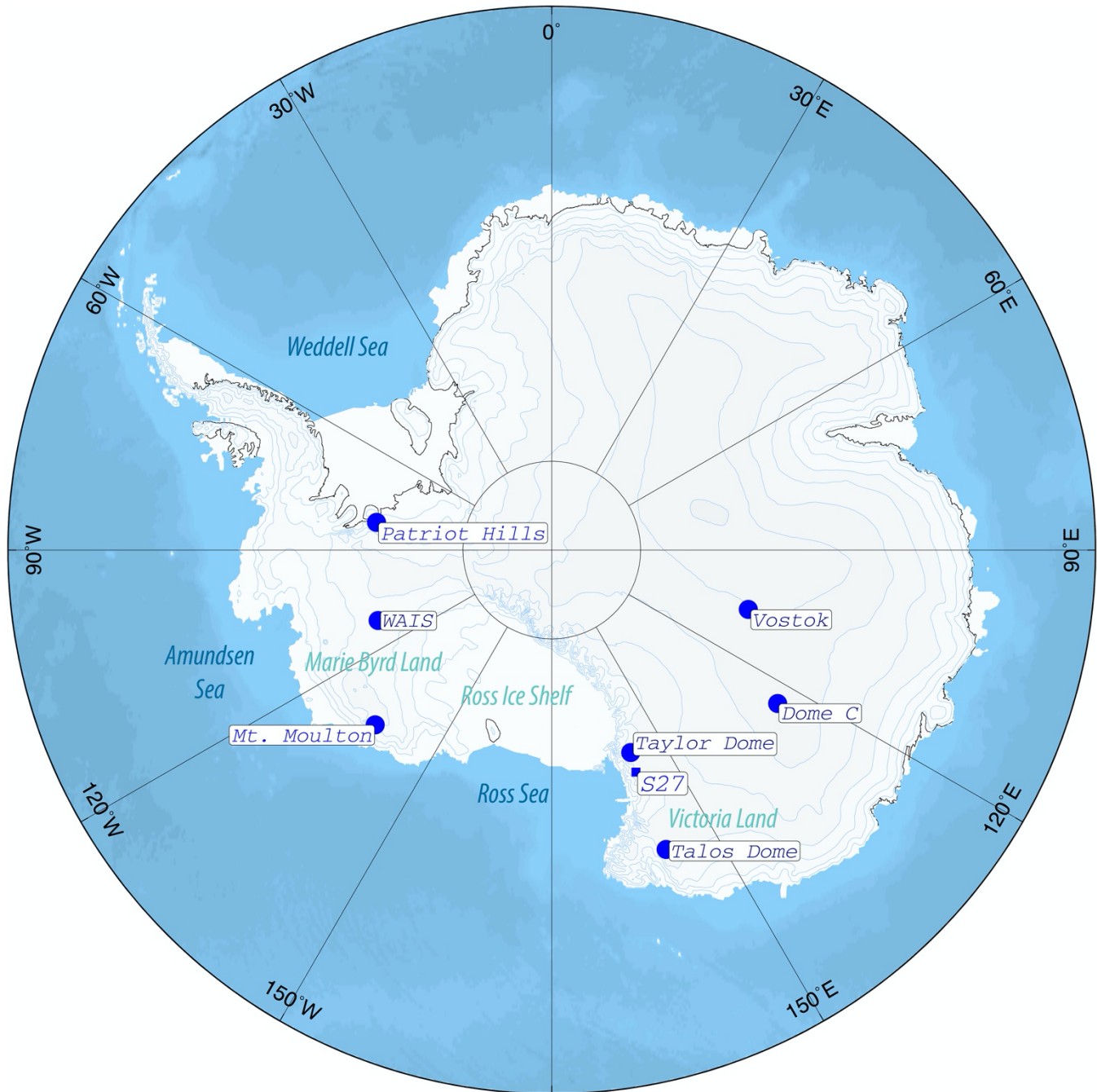

**Figure 1: Locations of key ice coring sites and geographic features of Antarctica.** The square indicates the location of Site 27 (S27). Other ice coring sites mentioned in this study are marked with blue circles (map source: Bindschadler et al, 2011).

The ice chronology of S27 was originally established by matching features in the stable water isotopes ($\delta D_{ice}$) to those in the EPICA Dome C (EDC) record (Figure 2; Spaulding et al, 2013). The $\delta$ notation here is defined as $[(R_{sample}/R_{standard}) - 1] \times 1000$ ‰, where $R$ is the raw ratio. Similarities between S27 and EDC $\delta D_{ice}$ overall give us confidence in the stratigraphic

continuity of the S27 ice core. By contrast, a preliminary gas age timescale is available in Spaulding et al (2013), constructed
by matching the $\delta^{18}O$ of atmospheric $O_2$ ($\delta^{18}O_{atm}$) measured in S27 (sample $N$ = 39) to the $\delta^{18}O_{atm}$ record of the Vostok ice
core (Figure S2). This preliminary $\delta^{18}O_{atm}$ record, however, did not capture a $\delta^{18}O_{atm}$ peak between 170 and 190 ka because
either the measurement is too sparse or the S27 ice core might not be continuous after all. We note, however, that $\delta D_{ice}$ in
this interval aligns well with the $\delta D_{ice}$ record at Vostok, supporting the continuity of the S27 core.

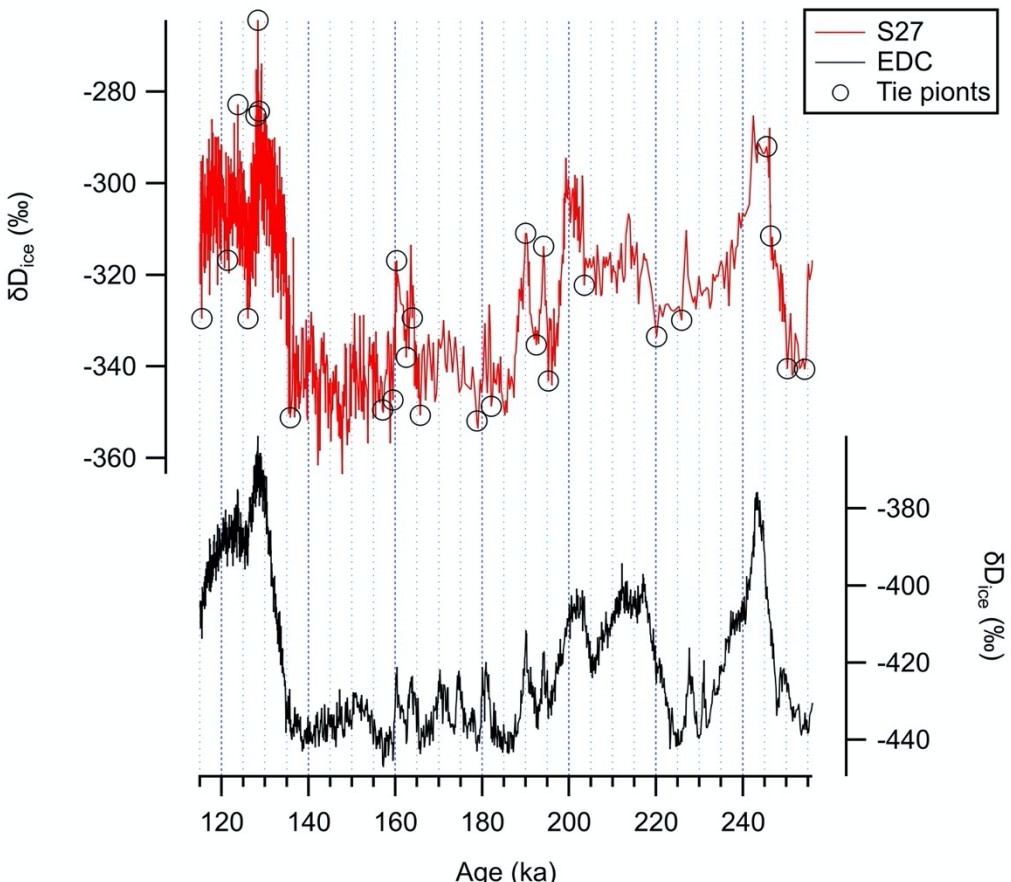

**Figure 2: Stable water isotope records in S27 (red; Spaulding et al, 2013) and EDC (black; Jouzel et al, 2007) between 115 and 255
ka.** Tie-points in S27 $\delta D_{ice}$ used by Spaulding et al (2013) are marked as circles.

In this study, we extend the existing S27 $\delta^{18}O_{atm}$ measurements by adding new $\delta^{18}O_{atm}$ values at 45 depths, including one that
overlaps with earlier data, in order to understand the stratigraphic integrity of the record at ~180 ka and further improve the
gas chronology. This collated $\delta^{18}O_{atm}$ record is then correlated with a recently published $\delta^{18}O_{atm}$ record of EDC (Extier et al,
2018) to derive a more accurate and complete gas chronology for S27. New measurements of $CH_4$ and $CO_2$ from the S27 ice
core are also used to further improve the $\delta^{18}O_{atm}$-based age scale between 105 and 147 ka. The gas chronology developed
here, together with the ice chronology reported in Spaulding et al (2013), yields the $\Delta$age, from which the accumulation rate

at Site 27 is estimated. We then examine the accumulation rate history in the context of atmospheric circulation changes and ice shelf/ice sheet stability during the LIG.

## 2 Material and methods

### 2.1 Glaciological setting

Site 27 (S27; 76.70°S, 159.31°E) ice core was drilled along the main ice flowline (MIF; Spaulding et al, 2012) of the Allan Hills region. It is situated to the northwest of the Convoy Range and the McMurdo Dry Valleys in Southern Victoria Land, East Antarctica, about 80 km from the Ross Sea coastline (Figure 1 and Figure S1). MIF ice flows slowly (<0.5 m yr$^{-1}$) from
90 the southwest to the northeast and feeds into the Mawson Glacier before eventually draining into the Ross Sea embayment. Horizontal ice velocities decrease as the ice approaches the Allan Hills nunatak, from 0.4-0.5 m yr$^{-1}$ in the upstream portion of the MIF to less than 0.3 m yr$^{-1}$ near where S27 is located, with the slowest ice flow rate in the area being 0.015 m yr$^{-1}$ (Spaulding et al, 2012).

The accumulation area of the ice feeding the Allan Hills BIA today lies about 20 km upstream (Kehrl et al, 2018). An
95 accumulation rate of 0.0075 m yr$^{-1}$ for the past ~660 years is inferred from a shallow firn core drilled near the Allan Hills BIA (Dadic et al, 2015). We regard this value as the present-day accumulation rate for the region where the blue ice at Allan Hills today was originally deposited. Note the accumulation rate of a blue ice record characterizes its original deposition site and is different from the surface mass balance within the blue ice field. Allan Hills BIA in particular is characterized by an ablation rate of 0.02 m yr$^{-1}$ (Spaulding et al, 2012). This negative mass balance leads to the exhumation of ice older than 100
100 ka at the surface (Spaulding et al, 2013).

### 2.2 δ$^{18}$O of O$_2$ (δ$^{18}$O$_{atm}$)

S27 δ$^{18}$O$_{atm}$ samples measured in this work share the analytical procedures described in Dreyfus et al (2007) and Emerson et al (1995) with several modifications. In brief, roughly 20 g of ice was cut from the core and the outer 2-3 mm trimmed away to prevent contamination from exchange with ambient air. The ice was then melted under vacuum to release the trapped air,
and the released gases were allowed to equilibrate with the meltwater for four hours (Emerson et al, 1995). After equilibration, the majority of the meltwater was discarded, and the remaining water refrozen at –30 °C. The headspace gases were subsequently collected cryogenically at –269 °C in a stainless-steel dip tube submerged in liquid helium. During the transfer to the dip tube, H$_2$O and CO$_2$ were removed by two traps in series, the first kept at –100 °C and the second placed inside a liquid nitrogen cold bath.

After gas extraction, the dip tube was warmed up to room temperature and attached to a dual-inlet isotope-ratio mass spectrometer (Thermo Finnigan Delta Plus XP) for elemental and isotopic analysis. δ$^{15}$N of N$_2$, δ$^{18}$O of O$_2$, and δO$_2$/N$_2$ were

measured simultaneously. All raw ratios were corrected for pressure imbalance between sample and reference sides of the mass spectrometer (Sowers et al, 1989). Pressure-corrected $\delta^{15}N$ of $N_2$ and $\delta^{18}O$ of $O_2$ were further corrected for the elemental composition of the $O_2$-$N_2$ mixture of the sample relative to the reference (Sowers et al, 1989). Next, $\delta O_2/N_2$ and $\delta^{18}O$ of $O_2$ were normalized to the modern atmosphere and corrected for gravitational fractionation that enriches the heavy molecules in the ice using $\delta^{15}N$ (Craig et al, 1988):

$$\delta O_2/N_{2,\,grav} = \delta O_2/N_2 - 4 \times \delta^{15}N \ldots\ldots\ldots\ldots\ldots\ldots\ldots\ldots\ldots\ldots\ldots\ldots\ldots\ldots\ldots (1)$$

$$\delta^{18}O_{grav} = \delta^{18}O - 2 \times \delta^{15}N \ldots\ldots\ldots\ldots\ldots\ldots\ldots\ldots\ldots\ldots\ldots\ldots\ldots\ldots\ldots (2)$$

The gravitationally corrected $\delta^{18}O$ is reported as $\delta^{18}O_{grav}$.

$\delta^{18}O_{grav}$ is frequently equal to $\delta^{18}O_{atm}$, the $\delta^{18}O$ of atmospheric $O_2$. In the case of this study, however, it is necessary to make an additional correction for post-coring gas losses using $\delta O_2/N_2$. Gas losses would lower $\delta O_2/N_2$ and elevate the $\delta^{18}O$ of $O_2$ trapped in ice and can occur in ice cores stored at or above –50 °C for an extended period of time (Dreyfus et al, 2007; Suwa and Bender, 2008). $\delta^{18}O$ of $O_2$ would also be elevated in ice that is extensively fractured (Severinghaus et al, 2009). In S27, $\delta O_2/N_2$ values measured five years apart clearly display the impact of gas losses, both in fractured and non-fractured ice (Figure S3).

In order to quantitatively correct for gas loss fractionations, we made the following assumptions: (1) $\delta^{18}O_{atm}$ samples measured in Spaulding et al (2013) have no gas loss and their $\delta O_2/N_2$ represents the true *in situ* value; (2) the systematic difference between the $\delta O_2/N_2$ values of the new samples measured in this study and those measured five years earlier is solely due to gas loss; and (3) in both fractured and non-fractured ice, the sensitivity of $\delta^{18}O_{atm}$ to gas loss (registered in the $\delta O_2/N_2$ values) is the same.

Gas loss correction for S27 $\delta^{18}O_{atm}$ is given by:

$$\delta^{18}O_{atm} = \delta^{18}O_{grav} + b \times \Delta\delta O_2/N_2 \ldots\ldots\ldots\ldots\ldots\ldots\ldots\ldots\ldots\ldots\ldots\ldots\ldots\ldots (3)$$

where $b$ is the slope of the regression line between the $\delta^{18}O_{grav}$ replicate difference versus the $\delta O_2/N_2$ replicate difference observed in new samples measured in this study. Ice with and without fractures has a statistically indistinguishable slope between the $\delta^{18}O_{grav}$ replicate difference versus the $\delta O_2/N_2$ replicate difference (Figure S4). We therefore use a unified gas loss correction equation. Because all but one of the new S27 samples were measured on depths different from the earlier samples, $\Delta\delta O_2/N_2$ cannot be directly computed. We regressed the $\delta O_2/N_2$ values against depth for the new and earlier datasets (Figure S3). $\Delta\delta O_2/N_2$ in Equation (3) was then calculated from the predicted $\delta O_2/N_2$ values at the same depth from the two regression lines. The absolute magnitude of this gas loss is on the order of 0.020 ‰.

After all corrections are applied, the pooled standard deviation of $\delta^{15}N$ and $\delta^{18}O_{atm}$ of S27 samples measured in this work is 0.012 ‰ and 0.067 ‰ ($N = 45$), respectively. Combined with the $\delta^{15}N$ and $\delta^{18}O_{atm}$ data presented in Spaulding et al (2013), the resulting pooled standard deviation for all of the S27 $\delta^{15}N$ and $\delta^{18}O_{atm}$ measurements ($N = 83$) is 0.041 ‰ and 0.046 ‰, respectively. S27 $\delta^{15}N$, $\delta^{18}O_{atm}$, and $\delta O_2/N_2$ data are available in Supplementary Data Table 1.

### 2.3 CO$_2$ and CH$_4$

S27 $CH_4$ was analyzed using a melt refreeze method described by Mitchell et al (2013). In short, ~60-70 g of ice was cut, trimmed, melted under vacuum, and refrozen at about –70 °C. $CH_4$ concentrations in released air were measured with gas chromatography and referenced to air standards calibrated by NOAA GMD on the NOAA04 scale. Precision is generally better than ±4 ppb. $CO_2$ concentrations were measured using a dry extraction (crushing) method described by Ahn et al (2009). 8-15 g samples were crushed under vacuum and the sample air was condensed in a stainless-steel tube at 11 K. S27

$CO_2$ concentrations were measured after equilibration to room temperature using gas chromatography, referenced to air standards calibrated by NOAA GMD on the WMO scale. Due to the relatively large size of the methane samples, cracks and fractures sometimes cannot be avoided and lead to potential contamination of greenhouse gases in the ambient air. No $CO_2$ and $CH_4$ data from S27 are available below 151 m in S27 due to extensive cracks. $CO_2$ and $CH_4$ samples above this interval were also excluded for age synchronization purposes if fractures were found present. Whenever possible, samples were

processed and analyzed in replicate for each depth and results averaged to obtain final $CH_4$ or $CO_2$ concentrations. Only $CO_2$ and $CH_4$ samples with two or more replicates are reported in Supplementary Data Table 2 and used in this study.

### 2.4 Gas age synchronization

We used the EPICA Dome C (EDC) ice core to synchronize the gas records for S27 between 115 and 255 ka because EDC has a more recent, higher resolution $\delta^{18}O_{atm}$ record available (Extier et al, 2018). We note that the ice chronology of S27 is

160 also based upon EDC (Spaulding et al, 2013). In addition, many Vostok $\delta^{18}O_{atm}$ measurements reported by Suwa and Bender (2008) were made on samples stored at –20 °C. They showed appreciable gas losses compared to Vostok samples analyzed during earlier times. The EDC $\delta^{18}O_{atm}$ record, some of which were obtained from samples stored at –50 °C, should be much less affected by gas loss than the Vostok samples are.

Figure 3 is the logic diagram outlining the gas age synchronization processes consisting of three steps: (1) match the extrema

(either a peak or a trough) in the S27 $\delta^{18}O_{atm}$ records to the extrema in EDC $\delta^{18}O_{atm}$ records ("peak match"); (2) match the absolute values of the remaining S27 $\delta^{18}O_{atm}$ to those of $\delta^{18}O_{atm}$ in EDC ("direct match"); and (3) if no match is available from either step (1) or (2), assign age by linearly interpolating the ages of their adjacent $\delta^{18}O_{atm}$ points.

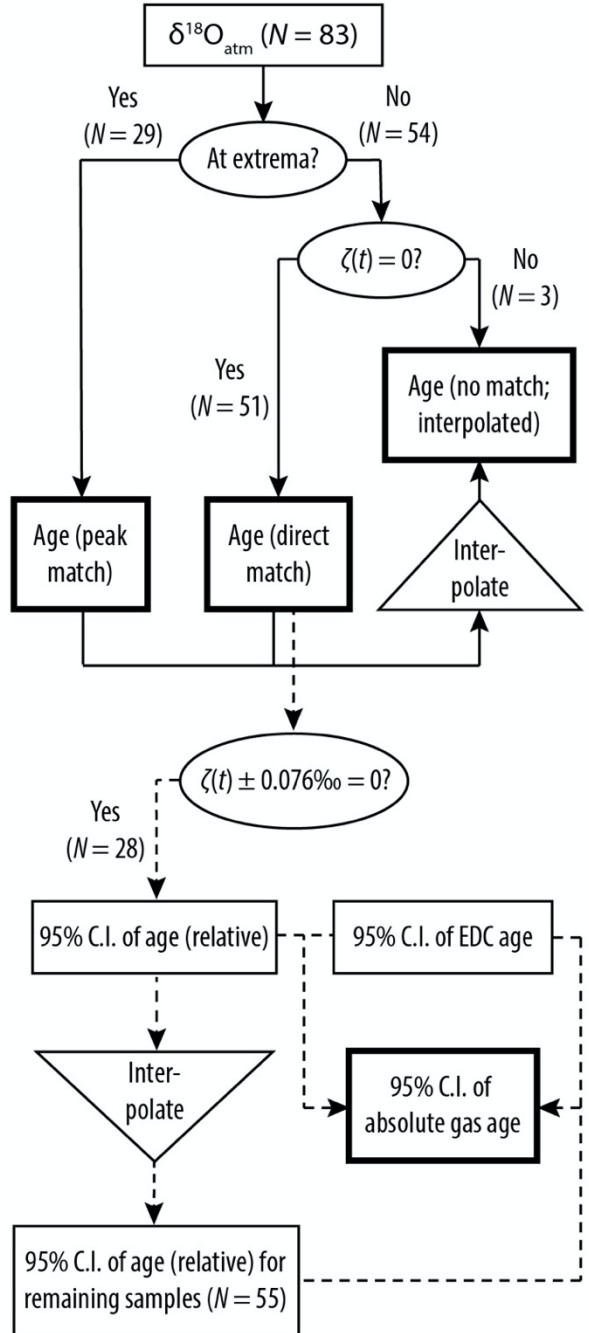

**Figure 3: Schematic workflow of timescale synchronization by $\delta^{18}O_{atm}$ (solid lines) and uncertainty estimates (dashed lines).** Rectangles refer to data, circles include conditional statements, and triangles stand for mathematical operations. Arrows mark the workflow. Boxes in bold lines indicate that we have arrived at the final answer. Definition of $\zeta(t)$ is given by Equation (4) in the text. C.I. = Confidence Interval.

In the first step, an extreme is defined when the $\delta^{18}O_{atm}$ sample is higher ("peak") or lower ("trough") than the two adjacent $\delta^{18}O_{atm}$ samples. The advantage of this approach is that it relies on the prominent features within the $\delta^{18}O_{atm}$ records and is not very sensitive to the systematic offset (if any) between the records. Out of 83 $\delta^{18}O_{atm}$ samples from S27, 29 (35 %) were identified as peaks/troughs and matched to corresponding features in the EDC $\delta^{18}O_{atm}$ record.

However, not all points are at peaks or troughs. To maximally utilize the rest of the data, we proceed with step (2) and constructed $\zeta$, a function of time $t$, defined below:

$$\zeta(t) = \delta^{18}O_{atm(t), S27} - \delta^{18}O_{atm(t), EDC} \dots\dots\dots\dots\dots\dots\dots\dots\dots\dots\dots\dots\dots\dots\dots (4)$$

Note $\delta^{18}O_{atm(t), EDC}$ here is linearly interpolated between individual $\delta^{18}O_{atm}$ points reported in Extier et al (2018). We seek the age $t$ that satisfies $\zeta(t) = 0$, in which case a "direct match" is deemed successful and a corresponding EDC age assigned to the S27 sample. 51 samples (61 %) have their ages assigned this way.

Finally, if a $\delta^{18}O_{atm(t), S27}$ point is neither at a peak or trough nor successfully matched to EDC $\delta^{18}O_{atm}$, the age of this data point is constrained by the ages of its adjacent $\delta^{18}O_{atm}$ points, as in step (3). Only three points (4 %) fall into the final category. The age assignment method and result of each S27 $\delta^{18}O$ datum are listed in Supplementary Data Table 3, along with their uncertainties. The final reported uncertainties associated with the gas chronology consists of three parts: the analytical uncertainties of $\delta^{18}O_{atm}$, the relative uncertainties of S27 chronology to EDC chronology, and the inherent uncertainties of the EDC chronology itself. Readers are referred to the Supplement for a more detailed discussion.

## 2.5 Firn densification inverse modeling

Firn densification models typically use accumulation rate and mean annual surface temperature to calculate $\Delta$age (see Lundin et al, 2017 for a more in-depth review). In our case, however, we seek to do the opposite, using the temperatures inferred from the isotopic composition of the ice $\delta D_{ice}$, and the $\Delta$age to estimate accumulation rates for S27. $\Delta$age is calculated by subtracting the gas age (obtained according to Section 2.4) from the ice age [from Spaulding et al (2013)] of the same depth.

Here, we employ an empirical firn densification model from Herron and Langway (1980), abbreviated as H-L hereafter. H-L has the merits of transparency and simplicity, as only three properties are involved. In any case, densification models are trained against similar data and simulate similar relations between temperature, accumulation rate, close-off depth, and close-off age ($\Delta$age). A limitation of empirical firn densification models is that their range of calibration may not include low-accumulation sites. To evaluate the performance of the H-L model, we compare the model output with present-day accumulation rate in the vicinity of S27 from Dadic et al (2015).

The H-L model divides firn densification into two stages. In the first stage (firn density < 550 kg m$^{-3}$), the densification process is independent of accumulation rate and is a function of surface temperature. At the threshold density of 550 kg m$^{-3}$, the elapsed time since snow deposition on the surface, $t_{0.55}$, is given by:

$$t_{0.55} = \frac{1}{k_0 \times A} \times \ln \left(\frac{\rho_i - \rho_0}{\rho_i - 0.55}\right) \dotfill (5)$$

$k_0$ is a temperature-dependent rate constant [$k_0 = 11 * e^{(-10160/8.314/T)}$], in which $T$ equals temperature, in Kelvin, to be inferred from $\delta D_{ice}$), $A$ is the accumulation rate (m yr$^{-1}$), $\rho_i$ is the density of ice (917 kg m$^{-3}$), and $\rho_0$ is the density of surface snow, which we assume to be 360 kg m$^{-3}$ (Herron and Langway, 1980). In the second stage, $t$, the total elapsed time since snow deposition, is calculated from firn density ($\rho$) using the following relationship:

$$t = \frac{1}{k_1 \times A^{0.5}} \times \ln \left(\frac{\rho_i - 0.55}{\rho_i - \rho}\right) + t_{0.55} \dotfill (6)$$

where $k_1$ is another temperature-dependent rate constant [$k_1 = 575 * e^{(-21400/8.314/T)}$], and $\rho$ is the firn density at this depth (Herron and Langway 1980). A key step in the firn densification process is bubble close-off, at which point gases can no longer diffuse within the firn and become "locked" between ice grains. For S27, bubble close-off is assumed to occur when the firn density reaches 830 kg m$^{-3}$. At this density, $t$ in Equation (6) equals $\Delta$age.

An additional parameter needed to solve accumulation rate $A$ from Equation (5) and (6) is the site temperature, $T$. In order to 215    derive historic Site 27 temperatures, we use $\delta D_{ice}$ reported in Spaulding et al (2013) and a regional isotope-temperature sensitivity of 4.0 ‰ °C$^{-1}$ established at the nearby Taylor Dome (Steig et al, 2000). We acknowledge that this isotope-temperature relationship could change over time, but it is not well-constrained in Southern Victoria Land (Steig et al, 2000). A recent estimate of isotope-temperature sensitivity for Talos Dome located in Northern Victoria Land yields a slope of 7.0 ‰ °C$^{-1}$ (Buizert et al, 2021). Increasing the isotope-temperature sensitivity by 75 % to 7.0 ‰ °C$^{-1}$ reduces the accumulation 220    rate estimates by no more than 20 %. Main conclusions of this paper would remain unchanged.

Modern-day Allan Hills $\delta D_{ice}$ of -270‰ (Dadic et al, 2015) and mean annual temperature of –30 °C (Delisle and Sievers, 1991) are used to calculate past temperatures. We note that the Allan Hills surface snow $\delta D_{ice}$ value found by Dadic et al (2015) is -257‰, but we opted not to use this value for the following reasons. First, this $\delta D_{ice}$ value is ~30‰ heavier than the Last Interglacial $\delta D_{ice}$ observed in the S27 core (Figure 2), implying an unlikely 7.5 °C warming today compared to the LIG. 225    Second, the deuterium excess value of the same surface snow is negative (-5‰), which may have been the result of post-depositional processes (Dadic et al, 2015). Third, a gradient of $\delta D_{ice}$ and deuterium excess along depth is observed, reaching -270‰ and 1‰ at the depth of ~0.25 m, respectively. Below this depth, both $\delta D_{ice}$ and deuterium excess show little variability (Figure 2 in Dadic et al, 2015). We interpret these observations to indicate post-depositional alterations to the isotopic compositions of snow above 0.25 m, and therefore use the $\delta D_{ice}$ values below 0.25 m (-270‰) for the isotope-

230 temperature calibration. In any case, using the nominal surface $\delta D_{ice}$ value of -257‰ for calibration increases the accumulation rate estimates by less than 10%.

Finally, we ran a Monte-Carlo simulation for each single $\Delta$age datum with 100,000 iterations to derive the distribution of accumulation rate estimates given the $\Delta$age uncertainties (See Supplement for its derivations). The reported accumulation rate comes from the value with the highest number of occurrences (the mode) and its 95 % confidence interval is bracketed by the values at 2.5th- and 97.5th-percentile, respectively (Figure S5).

The H-L model also produces estimates of the depth at which firn density crosses the bubble close-off threshold. The interval from the close-off depth to the surface contains three components: the lock-in zone where ice layers are impenetrable and vertical transport is inhibited ($h_{LIZ}$); the height of the diffusive column where the gravitational separation of heavy isotopes occurs ($h_{diff}$); and the height of the convective zone where vigorous mixing by convective air motions prevents the
240 establishment of gravitational profiles ($h_{conv}$; Severinghaus et al, 2010).

The following barometric equation is used to link the diffusive column height $h_{diff}$ to the $\delta^{15}N$ enrichment (Sowers et al, 1989):

$$\delta^{15}N = e^{\frac{\Delta m \times g \times h_{diff}}{R \times T}} - 1 \dots\dots\dots\dots\dots\dots\dots\dots\dots\dots\dots\dots\dots\dots\dots\dots (6)$$

where $\Delta m$ is the difference between the molecular weight of $^{15}N^{14}N$ and $^{14}N^{14}N$ (0.001 kg mol$^{-1}$), $g$ is the gravitational
acceleration constant (9.8 m s$^{-2}$), $R$ is the ideal gas constant (8.314 J mol$^{-1}$ K$^{-1}$), and $T$ is the temperature (in Kelvin). $h_{diff}$ is calculated by subtracting the $h_{LIZ}$ (3 m) and $h_{conv}$ (0 m) from the bubble close-off depth calculated using the H-L model.

## 3 Results

### 3.1 A new gas chronology for S27

Figure 4 shows the result of synchronization between the S27 and EDC via $\delta^{18}O_{atm}$. Each of the $\delta^{18}O_{atm}$ minima and maxima
associated with orbital-scale insolation variations between 105 and 245 ka is successfully identified in S27, including the $\delta^{18}O_{atm}$ peak around 180 ka that was previously missing in Spaulding et al (2013). Overall, the strong similarities between the two $\delta^{18}O_{atm}$ series give confidence to the stratigraphic integrity of the S27 gas record. The match between the S27 and EDC $\delta^{18}O_{atm}$ is particularly tight between 105 and 145 ka, which corresponds to the depth interval of 7 m to 145 m in S27. $\delta^{18}O_{atm}$ samples older than 145 ka show more offsets between S27 and EDC. This is noticeably evidenced by the scattering
of S27 $\delta^{18}O_{atm}$ data around the EDC curve between 202 and 210 ka. One possible explanation for the increased scatter is a decline in core quality in S27, where ice below ~145 m is heavily fractured and visibly characterized by uneven surface cracks. This explanation is supported by more variable S27 $\delta O_2/N_2$ values below 150 m (Figure S3), suggesting a critical

point between 145 and 150 m, below which depth data reproducibility deteriorates substantially. This variability would be accompanied by more noise in the $\delta^{18}O_{atm}$ record despite the corrections for gas loss in the deeper part of the ice core.

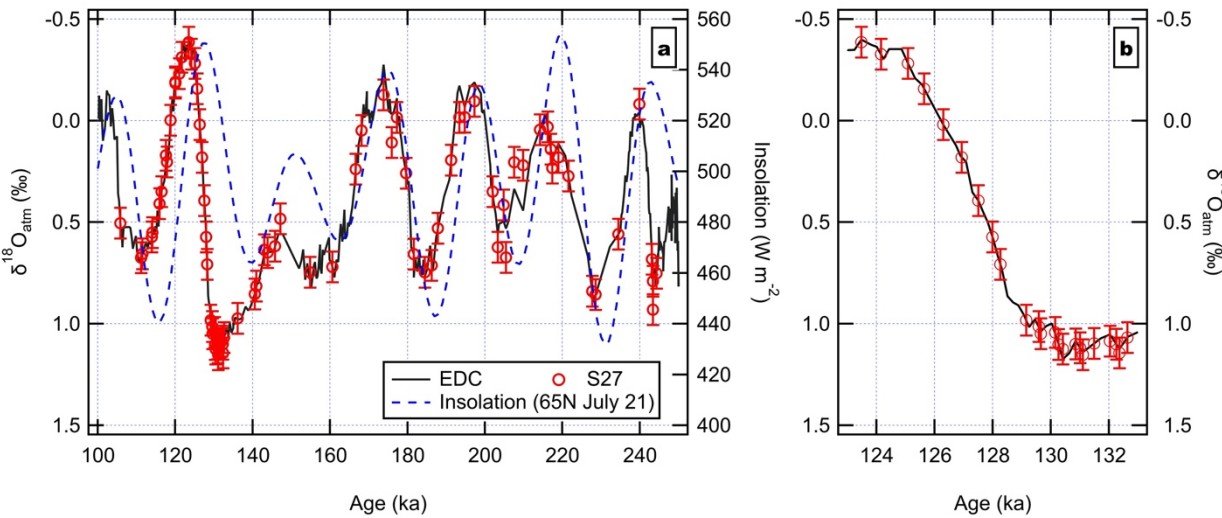

**Figure 4: $\delta^{18}O_{atm}$ measured in S27 (red) was matched to a high-resolution $\delta^{18}O_{atm}$ record from EDC (black) between 100 and 250 ka by Extier et al (2018): (a) the whole record between 114 and 255 ka, accompanied by the northern hemisphere (65 N) July 21st insolation curve (dashed blue) and (b) a close-up view between 123 and 133 ka.** S27 $\delta^{18}O_{atm}$ data include those reported in Spaulding et al (2013) and additional $\delta^{18}O_{atm}$ samples measured in this work. Error bars represent 95% confidence interval of the combined EDC and S27 $\delta^{18}O_{atm}$ measurements, following the approaches described in the Supplement.

CH4 ($N = 12$) and CO2 ($N = 17$) samples measured at S27 are plotted on the $\delta^{18}O_{atm}$-derived timescale in Figure 5. Also shown for comparison are an EDC CH4 (Loulergue et al, 2008) record and a composite high-resolution CO2 record built upon multiple Antarctic ice cores (Bereiter et al, 2015 and references therein). Because the relative uncertainties associated with greenhouse gas measurements are smaller than the errors of the $\delta^{18}O_{atm}$ analyses, greenhouse gases are used to further improve the $\delta^{18}O_{atm}$-derived gas chronology. Below we describe the process of fine-tuning the $\delta^{18}O_{atm}$-derived gas chronology to better match greenhouse gas measurements in the reference records. We also tabulate chosen tie-points between the S27 and EDC CH4, as well as S27 and the composite CO2, in Supplementary Data Table 2. The timescale adjustment below only applies to the interval between 115.7 and 147.2 ka: at 115.7 ka, both S27 CO2 and CH4 agree well with the co-eval values observed in the reference records, and S27 and EDC $\delta^{18}O_{atm}$ values were both extrema at 147.2 ka. They are selected as "anchor points" that do not involve any adjustment.

The most prominent feature in Figure 5 is the greenhouse gas peak at ~128 ka. There is a 2-ppm offset in this CO2 peak observed in the S27 record compared to the composite record (within the analytical uncertainty). There is an offset of only 163 years between the ages of the CO2 peaks recorded at S27 and in the composite record. We therefore tied the CO2 peak at 128.6 ka in S27 to the peak at 128.5 ka in reference time-series. In the ice below, both CH4 and CO2 in S27 show a clear increasing trend with time going upward towards the maximum, corresponding to the deglacial rise of greenhouse gases. We

tied the S27 CH₄ data point at 144.2 ka with the EDC CH₄ point at 144.8 ka. We acknowledge that the low sampling resolution of greenhouse gases below 140 m precludes more rigorous evaluation of the $\delta^{18}O_{atm}$-derived gas chronology.

Ages of the data points in between the anchor and tie points were re-sampled by linear interpolation. Uncertainties of the gas chronology are assumed to be unaffected by this fine-tuning. The new, complete gas chronology for S27 is presented in Supplementary Data Table 3. We emphasize the effect of this fine-tuning on the gas chronology is at most 600 years (at 144.2 ka), and in many cases much smaller. Even with the timescale solely derived from $\delta^{18}O_{atm}$, the conclusions of this study remain the same. The final uncertainty associated with the new S27 gas age scale (1σ) varies between ±1600 and ±4000 years, averaging ±2200 years, also listed in Supplementary Data Table 3.

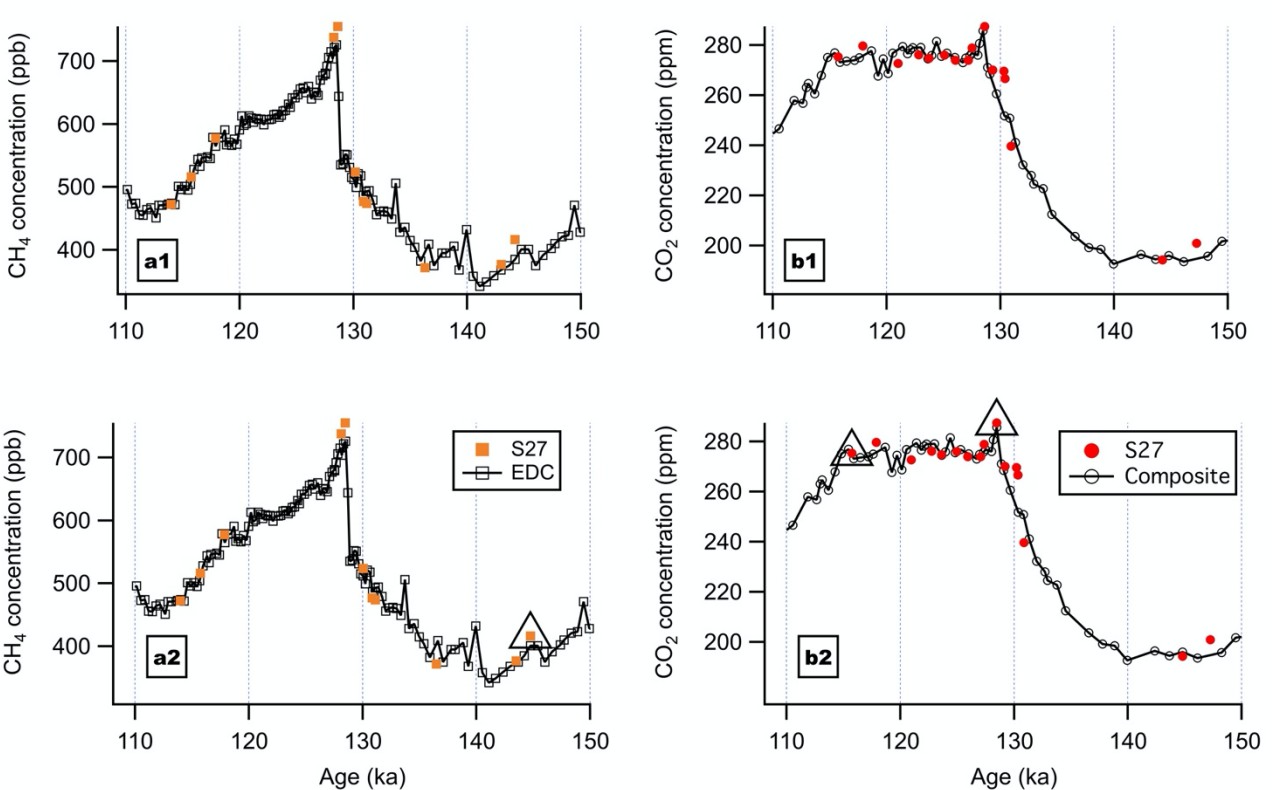

Figure 5: CH₄ and CO₂ measured in S27 plotted on the timescale developed solely on $\delta^{18}O_{atm}$ (a1 and b1) and on the chronology after greenhouse gas synchronization (a2 and b2). Tie points and anchor points are marked in triangles. EDC CH₄ data are from Loulergue et al (2008), composite CO₂ data from multiple ice cores are from Bereiter et al (2015) and references therein, and the timescale (AICC2012) is from Veres et al (2013) and Bazin et al (2013).

### 3.2 Ice age-gas age difference (Δage)

Below we evaluate Δage calculated by subtracting gas age from ice age. Gas age comes from the $\delta^{18}O_{atm}$-derived, CH₄- and CO₂-adjusted gas chronology from this work. Ice age comes from the $\delta D_{ice}$-based ice chronology established in Spaulding et

al (2013). All chronologies discussed here have been converted to AICC2012, the most up-to-date Antarctic ice core timescale (Veres et al, 2013; Bazin et al, 2013).

The relationship between the depth and the ice and gas age in S27 is shown in Figure 6. The ice age is younger than the gas age between 192 and 204 m. This result is glaciologically impossible given the presence of a diffusive column and the measured positive $\delta^{15}N$ values (Figure S6). Such discrepancies could arise from the ambiguous matching of $\delta D_{ice}$, severe impact of gas losses on $\delta^{18}O_{atm}$, or both.

The interval between 115 and 140 ka in S27 is where the gas and ice age scales are both well-constrained, corresponding to a depth range of 10.05 and 134.55 m. Selection of $\delta^{18}O_{atm}$ samples from this section was done so that there are no visible fractures and cracks, and we therefore limit our subsequent discussion of Δage to the interval between 115 and 140 ka. Here, $\delta D_{ice}$ values represent deglacial warming, and the cooling after the LIG, allowing unambiguous feature matching (e.g. the distinct LIG peak around 128.2 ka). We acknowledge that different ice core water isotope records may not be perfectly synchronous and it is possible that the selected tie-points do not capture the true peaks (or troughs) due to the discrete sampling of the ice cores. These errors are incorporated in the form of relative uncertainty in the ice age scale and propagated along with the intrinsic uncertainty associated with EDC Δage on AICC2012 timescale into the S27 Δage uncertainty. We discuss those considerations in greater detail in the Supplement.

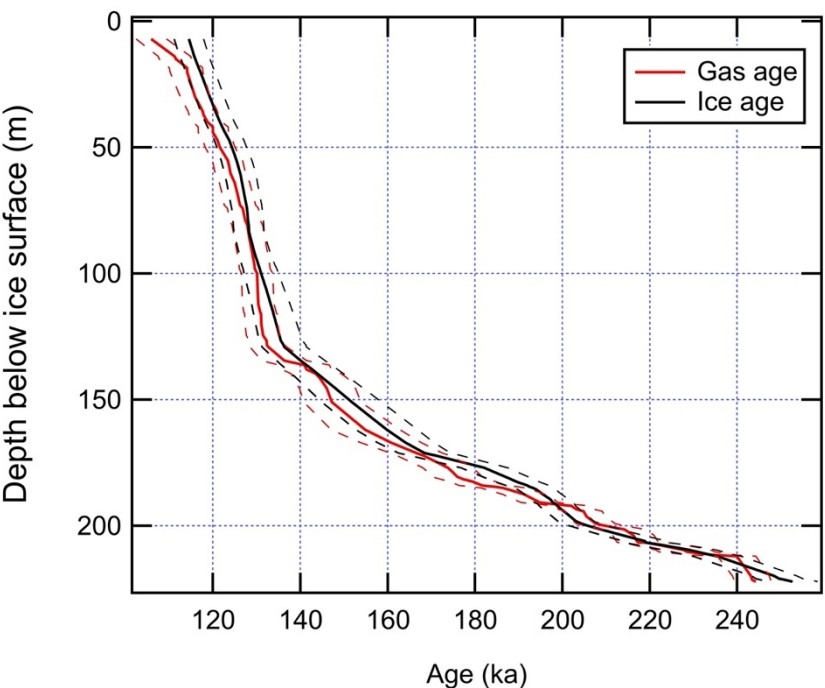

**Figure 6: Depth profile of gas age (red) and ice age (black) in S27.** Dashed lines represent the 95% uncertainty of the absolute age.

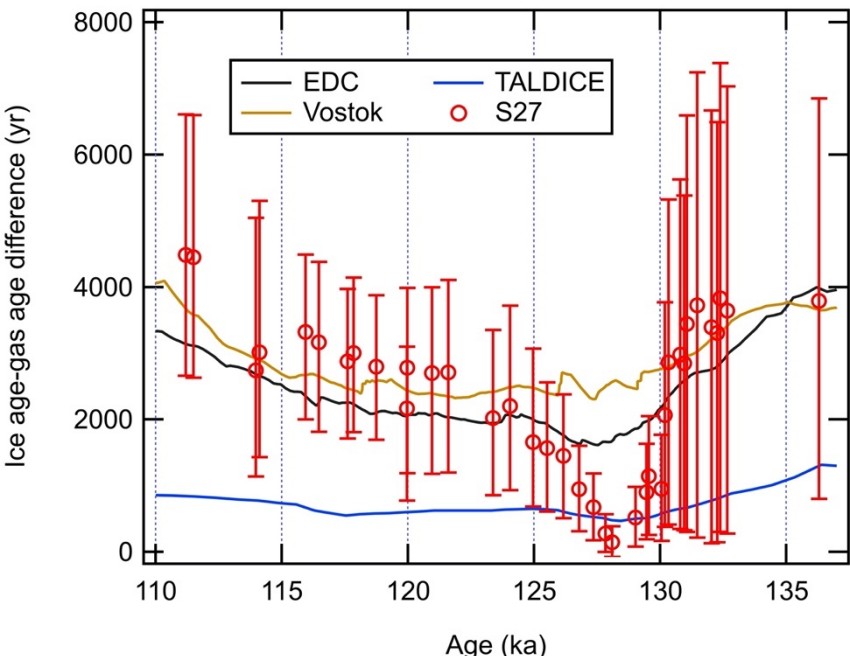

**Figure 7: Ice age-gas age difference (Δage) between 110 and 136 ka in S27 (red), Talos Dome (TALDICE; blue), Vostok (brown),**
**and EDC (black).** This age interval corresponds to the depth range of 10.05 to 134.55 m in S27, where the measured samples are free from any visible fractures. TALDICE, Vostok, and EDC Δage data are from Bazin et al (2013) and references therein. All four Δage records show similar pattern during the Termination II, with minimum values reached around ~128 ka. Error bars in S27 data represent the 95% confidence interval of the Δage estimates. Note that the Δage values are plotted on the gas age scale and the confidence intervals of
the S27 Δage points are asymmetrical.

Δage of S27 between 115 and 140 ka is plotted along with the Δage estimates in Talos Dome, EDC, and Vostok (Figure 7). Apart from the similarities in the shape of the Δage curve across Termination II between the four records, a prominent feature here is the very low Δage of S27 during the LIG, reaching its minimum value of 145 years (95 % CI: 27-300 years) at 128.2 ka. In the entire record, small Δage values are not limited to 128 ka. For example, the Δage reaches less than 500 years

at ~142 ka, but we caution that the interval between 140 and 145 ka is under-constrained. Between the interval of 140 and 145 ka, there are only two ice age tie-points (Figure 2) and four $\delta^{18}O_{atm}$ measurements (Figure 4). The very small Δage around 142 ka cannot be established as a robust feature. By contrast, there are 15 $\delta^{18}O_{atm}$ samples in the 5,000-year interval from 128 to 133 ka and four ice age tie-points within the interval between 125 and 130 ka.

Finally, we acknowledge that given the noisy nature of the S27 $\delta D_{ice}$ records (Figure 2 and Figure S7), it is possible that the
Δage—and by inference the ice accumulation rates—could have larger errors than reported here, which we discuss in greater detail in the Supplement. A more refined ice chronology, perhaps made available by absolute-dating tephra layers and synchronizing ion content such as sulfate, will greatly improve the ice chronology of S27. However, the Allan Hills volcanological records are dominated by regional tephra, with similar composition and contamination by plagioclase crystals from the basement bedrock. In any case, past efforts to date tephra layers directly using Ar method were not successful

(William McIntosh, personal communications). Abundant regional volcanic signals further complicate direct correlation of Allan Hills volcanic record with deep Antarctic ice cores (Nishio et al, 1985).

### 3.3 Accumulation rates

Figure 8 shows the estimated accumulation rate in S27 versus time between 115 and 140 ka, determined from estimates of $\Delta$age [i.e. solving $A$ from $t$ in Equation (6)]. Accumulation rates in three other aforementioned Antarctic ice cores (Talos
Dome, EDC, and Vostok) during the same time interval are shown for comparison. While the accumulation rate in other sites began to increase around 136 ka, coinciding with the onset of Termination II, the accumulation rate at S27 remained low for another ~3,000 years, averaging 0.0026 m yr$^{-1}$ from 140.1 to 133.2 ka.

Beginning at 132.2 ka, S27 accumulation rate increased by an order of magnitude within 4,000 years and reached its maximum value at 0.086 m yr$^{-1}$ (95 % CI: 0.059~0.904 m yr$^{-1}$) at 128.2 ka. The peak in S27 accumulation rates coincides
with ~128 ka peak warming in Antarctica, as well as with the maximum accumulation rate recorded in three other East Antarctic ice cores. We acknowledge the large uncertainty here, as high accumulation rate estimates are associated with a very small $\Delta$age values and hence large relative errors. That said, this particular small $\Delta$age is a robust estimate, because the precise match between $\delta D_{ice}$ peaks around ~128 ka puts a firm constraint on ice age (Figure 2), and the monotonic deglacial $\delta^{18}O_{atm}$ change means small gas age uncertainty (Figure 4b). In addition, this estimate agrees with the peak LIG
accumulation rate at the nearby Taylor Dome deduced from $^{10}$Be activities in the ice (0.074 m yr$^{-1}$; Steig et al, 2000). Importantly, even the most conservative accumulation rate estimate of 0.059 m yr$^{-1}$ (the lower bound of the 95 % CI) still means an order-of-magnitude increase in the LIG S27 accumulation rate.

The elevated snow accumulation during the LIG at S27 was a transient phenomenon as by 125.5 ka, accumulation rates had already dropped below 0.075 m yr$^{-1}$, the present-day accumulation rate observed in the vicinity of Allan Hills (Dadic et al,
2015), and further declined to a baseline value of less than 0.005 m yr$^{-1}$ after 120 ka. The H-L model in general produce estimates for $\delta^{15}$N values that agree with the observations, except between 125 and 130 ka where the H-L model appears to overestimate $\delta^{15}$N (Figure S6). This could be explained by the presence of a convective column (Severinghaus et al, 2010). We note, however, that the occurrence of deep convection in the firn column does not impact accumulation rate estimates from $\Delta$age because we are not relying on $\delta^{15}$N values to reconstruct lock-in depths.

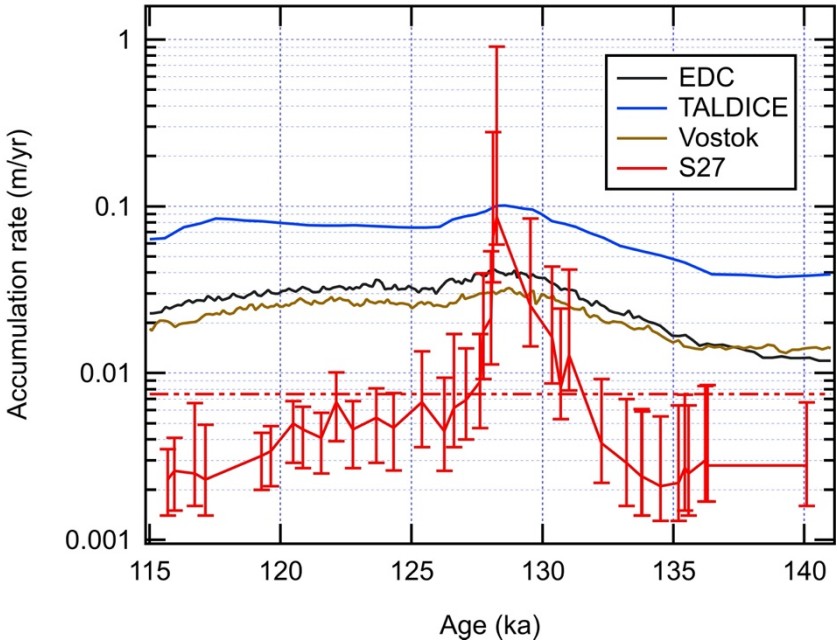

**Figure 8: Accumulation rate between 115 and 140 ka in S27 (red), Talos Dome (TALDICE; blue), Vostok (brown), and EDC (black).** Note the y-axis is plotted on logarithm scales. Accumulation rates of EDC, TALDICE, and Vostok are from Bazin et al (2013) and the references therein. Error bars represent the 95% confidence interval for S27 accumulation rate estimates. The dashed line in red represents the present-day accumulation rate (0.0075 m yr[-1]) in the vicinity of Allan Hills (Dadic et al, 2015).

## 4 Discussion

Today, moisture transport into Allan Hills vicinity is primarily in the form of synoptic-scale low-pressure weather systems (Cohen et al, 2013) modulated by the position and intensity of the Amundsen Sea Low and the austral westerlies (Bertler et al, 2004; Patterson et al, 2005). In this context, one way to interpret the pronounced increase in S27 accumulation rates during Termination II is a transient reorganization of large-scale atmospheric circulation due to the poleward shift in the westerly wind belt associated with the deglacial warming. This mechanism is commonly invoked to explain the $CO_2$ rise during ice age terminations (Toggweiler et al, 2006). Simulations also reveal increased precipitation in Southern Victoria Land by the end of 21st century due to enhanced moisture transport towards the interior of the continent in a warmer climate state (Krinner et al, 2007). Atmospheric $CO_2$ during Termination II began to increase around 140 ka and peaked around 128.5 ka, coinciding with the accumulation rate peak in S27 (Figure 9). The contraction of the westerlies would push the storm tracks further into the Antarctic continent. The results would be increased precipitation at otherwise low-accumulation sites, and the peak in accumulation concomitant with the maximum atmospheric $CO_2$.

In addition to large-scale circulation shifts, local to regional changes must also be at work for two reasons. First, peak accumulation rate at S27 during the LIG is an order of magnitude larger than the average S27 accumulation rate between 140

and 133 ka. This difference is at least three times larger than the doubling or tripling in accumulation rates recorded in other Antarctic ice cores (Figure 8). Second, the S27 accumulation rate started to increase at 132 ka and lagged the Antarctic warming and $CO_2$ increase (Figure 9). The abrupt change in accumulation rates could be possibly linked to the migration of local ice domes and the subsequent shifts in accumulation gradient, as revealed by some pioneering studies in this region (Morse et al, 1998; Morse et al, 1999). More recently, studies comparing the Taylor Dome and Taylor Glacier blue ice accumulation rates find a reversal in the accumulation gradient in the Last Glacial Maximum compared to Marine Isotope Stage 4 without glacial-interglacial transitions (Baggenstos et al, 2018; Menking et al, 2019). We cannot fully rule out this possibility. However, we note that the peak LIG accumulation rate estimated from S27 is comparable to the Taylor Dome ice record (0.074 m yr$^{-1}$; Steig et al, 2000) and Talos Dome (0.101 m yr$^{-1}$; Bazin et al, 2013). It appears that the high accumulation rates observed in the LIG are a regional signal in Victoria Land. We thus proceed with the interpretation that the increase in accumulation rate during MIS 5e reflects a regional climatic shift. A more rigorous test would be extending the accumulation history beyond 140 ka and seek large accumulation increase in glacial intervals.

We hypothesize that the peak S27 accumulation rate at 128 ka may reflect more open-ocean conditions in the Ross Sea. More open-water conditions near S27 could result from (1) reduced sea ice extent and an increase in polynya size, and/or (2) retreat of the Ross Ice Shelf. Our first hypothesis concerning sea ice extent is supported by the blue ice record from the Mt Moulton BIA (76° 04'S, 134° 42'W; Figure 1) near the Ross Sea coast in West Antarctica. The Mt Moulton record clearly documents an increase in sea salt concentrations and the lowest level of non-sea-salt sulfate during the peak LIG warming, interpreted as a minimum extent of sea ice and therefore the proximity of Mt Moulton ice field to an open ocean at 128 ka (Korotkikh et al, 2011). Moreover, Holloway et al (2016) demonstrate that the retreat of winter sea ice in the Southern Ocean is fully capable of explaining the distinctive 128 ka $\delta^{18}O$ isotope peak observed in Antarctic ice cores, although it should be noted that the inferred sea ice retreat in the Ross Sea is minimal in Holloway et al (2016). If our hypothesis of more open-water conditions in the Ross Sea is correct, a LIG spike in sea salt concentration similar to that in the Mt Moulton ice record should also be visible in the S27 record. Furthermore, as moisture originating from higher latitude generally has lower deuterium excess values, the open-water conditions at the peak of LIG would lower the deuterium excess in the S27 ice. We note, however, that no aerosol or deuterium excess data from S27 are available at present.

The second hypothesis concerns the Ross Ice Shelf (RIS). Since the Last Glacial Maximum (LGM) the RIS has shrunk in size (Ship et al, 1999; Yokoyama et al, 2016), and recent work on glacial deposits in the southern Transantarctic Mountains reveals rapid grounding line retreat in the central and western Ross Sea in the early Holocene (Spector et al, 2017). Morse et al (1998) first proposed that the elevated topography and the expansion of RIS during the LGM drove the storms heading towards Victoria Land northward, supported by later studies such as Aarons et al (2016). If the RIS has been capable of exerting influence on the synoptic weather systems over the last 13,000 years, the same underlying mechanism could also be operating in the LIG. That is, a further retreat of RIS led to the southward displacements of storm tracks and the enhancement of moisture transport into Site 27's accumulation region. Indeed, surface airflow into Victoria Land via the

Ross Sea is enhanced in some numerical simulations where WAIS and its adjacent ice shelves are removed (Steig et al, 2015). McKay et al (2012) in addition suggest the absence of ice shelf cover in the western Ross Sea sometime in the past 250 thousand years, a scenario compatible with the LIG retreat of RIS inferred from the S27 accumulation rate record. We

acknowledge that the hypothesized response of atmospheric circulation to the absence of the RIS and reduced sea ice extent requires more examination by climate models.

The reduced ice shelf extent in the LIG has important implications for the stability of the West Antarctic Ice Sheet, as a widespread ice shelf retreat could signify the breakup of continental ice masses (DeConto and Pollard, 2016; Garbe et al, 2020). The inference of increased open ocean water in the Ross Sea during the LIG aligns with other observations. Several

lines of geologic evidence point to a rising sea level in the early LIG between 129 and 125 ka, implying substantial mass losses from continental ice sheets (e.g. McCulloch and Esat, 2000; O'Leary et al, 2013; Dutton et al, 2015b). Temperature reconstructions for North Greenland show similar-to-modern values at 129 ka (NEEM community members, 2013) and suggest that the Greenland Ice Sheet (GIS) was not responsible for the sea-level high stand at that time (Yau et al, 2016). In this case, the elevated sea level beginning ~129 ka would have to significantly result from mass losses in West Antarctica,

according to a recent LIG sea-level reconstruction with high temporal resolution and precise chronological controls (Figure 9; Rohling et al, 2019). This reconstruction is further reinforced by basin-wide ice losses in the Weddell Sea as early as 129 ka deduced from a blue ice record from Patriot Hills, West Antarctica (80° 18′S, 81° 21′W; Figure 1; Turney et al, 2020).

An open Ross Sea at 128 ka inferred from the S27 ice record presented in this work supports the proposed early collapse of the WAIS in the LIG, and underscores the vulnerability of Antarctic ice shelves and ice sheets to rising ocean temperatures

during Termination II, possibly linked to Heinrich event 11 between 135 and 130 ka (Figure 9; Marino et al, 2015). For example, a chain of events could be that meltwater discharge in the North Atlantic weakened the Atlantic Meridional Overturning Circulation and consequently reduced the northward cross-equatorial heat transport, resulting in Southern Hemisphere warming, southward shifts of the intensified westerlies, and enhanced $CO_2$ ventilation in the Southern Ocean (Menviel et al, 2018). Even if the S27 accumulation rate increase since 132 ka was not accompanied by the collapse of the

WAIS in the early LIG, our results appear at odds with some model predictions that the collapse of WAIS only began at 127 ka while the RIS remained largely intact (e.g. Clark et al, 2020). Based on the accumulation rate history, the Ross Ice Shelf appeared to quickly readvance soon after the conclusion of peak warming. By 125 ka the open oceans conducive to high accumulation rate at S27's accumulation region became reverted to its previous conditions covered by ice. Since it is unlikely that the WAIS could recover within 3,000 years, this recovery likely stems from the fact that the Ross Ice Shelf is

partly fed by the East Antarctic Ice Sheet (Rignot et al, 2011), which remained stable during the LIG.

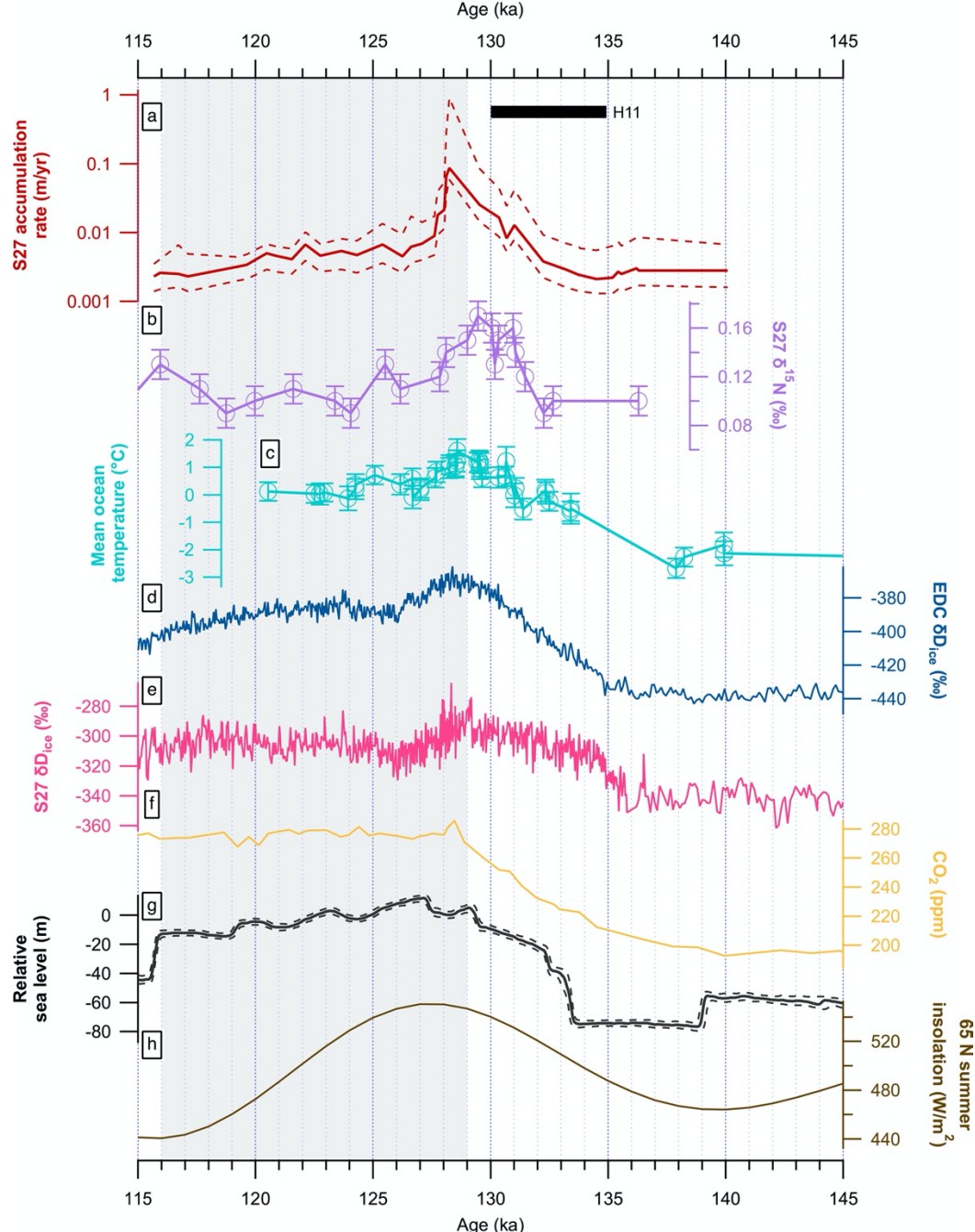

**Figure 9: Paleoclimate records during Termination II and the Last Interglacial.** (a) S27 accumulation rate (this study) deduced from ice without any visible fractures. (b) $\delta^{15}N$ of $N_2$ in S27 (this study). (c) Mean ocean temperature based on noble gas ratios (Shackleton et al, 2020). (d) S27 stable water isotope records (Spaulding et al, 2013). (e) EPICA Dome C $\delta D_{ice}$ records (Jouzel et al, 2007). (f) Atmospheric $CO_2$ (Bereiter et al, 2015 and references therein). (g) Sea-level relative to present day inferred from $\delta^{18}O$ of Red Sea planktonic foraminifera with 95% confidence interval marked by the dashed lines (Rohling et al, 2019). (h) July 21st insolation at 65 N. Shaded zone marks the LIG. The black bar marks Heinrich event 11 (H11).

Finally, regardless of the cause(s) of the LIG spike in S27 accumulation rate, it has significant glaciological implications in terms of ice flow modeling for the Allan Hills BIA, where ice older than 2 million years (Ma) has been discovered in sites disconnected from the main ice flow line (Yan et al, 2019). Recent ice-penetrating radar surveys and ice flow modeling by Kehrl et al (2018) have suggested the potential preservation of a stratigraphically continuous ice record, with 1 Ma ice located 25 to 35 m above the bedrock. The modeling efforts by Kehrl et al (2018) assume no-higher-than-present accumulation rates in the past and constant sublimation rates that control the exhumation of ice along the flow line. In light of the discovery of this work, models with the time-dependent accumulation rate constrained by observations could better predict the age-depth profile. Nonetheless, S27 itself provides a continuous, readily available ice record for Termination II and the LIG with a nominal resolution of 285.7 yr m$^{-1}$ in the upper 145 m with good ice core quality, making the Allan Hills BIA an appealing archive for paleoclimate investigations targeting the LIG.

**5 Conclusion**

We present an improved gas chronology for a shallow blue ice record (S27) drilled in the Allan Hills Blue Ice Area, East Antarctica, located in close proximity to the current northwest margin of the Ross Ice Shelf (RIS). The new S27 gas chronology is derived from the $\delta^{18}O$ of $O_2$ trapped in the ice. Complementary $CH_4$ and $CO_2$ measurements validate and refine the gas chronology, paving the way for future utilization of S27 samples. Calculation of accumulation rate on the basis of the ice age-gas age differences between 115 and 140 ka in S27 reveals a dramatic increase in accumulation rate since 132 ka and peaking at 128.2 ka, coinciding with the peak LIG Antarctic warming and atmospheric $CO_2$.

We hypothesize that in addition to changes in the large-scale atmospheric circulation affecting precipitation on the Antarctic continent, sea ice and ice shelf extent could alter local meteorological boundary conditions and lead to the observed spike in accumulation rate. A greater reduction in size of the RIS would cause the storm tracks that bring substantial precipitation to Victoria Land today to migrate farther south. The ice shelf retreat would also be compatible with a high sea-level stand around 129 ka sourced from the collapse of the West Antarctica Ice Sheet near the onset of Last Interglacial period (Yau et al, 2016; Rohling et al, 2019). If this was the case, an early collapse of WAIS (along with the RIS) in the LIG would underscore its vulnerability to rising temperatures.

Our data suggests that, soon after the conclusion of peak warming, the open ocean conducive to high accumulation rate near S27's accumulation region became once again covered by ice by 125 ka. The depositional site of S27 returned to its previous conditions characterized by low accumulation rates similar to those today. We conclude that if the Ross Ice Shelf indeed collapsed early in the LIG, it would have quickly re-advanced by 125 ka, possibly fed by the ice streams sourced from the East Antarctic Ice Sheet.

**Data availability**

Data supporting the conclusions of this paper are available in the supplement of the manuscript (Supplementary Data Table 1-4). In addition, S27 ice core data underlying this study will be made publicly available on the United States Antarctic Program Data Center (http://www.usap-dc.org/) upon the acceptance of this manuscript for publication with the following Digital Object Identifiers (DOIs): 10.7265/N5NP22DF (S27 stable water isotope records); 10.15784/601424 (S27 gas isotopes); and 10.15784/601425 (S27 greenhouse gas concentrations).

**Author contribution**

Y.Y. conceptually conceived the study. Y.Y., M.L.B., E.J.B., A.V.K., and P.A.M. designed the experiments. Y.Y. performed the new $\delta^{18}O_{atm}$ analyses. J.A.H. undertook earlier $\delta^{18}O_{atm}$ measurements. N.E.S. measured $\delta D_{ice}$ and established the ice chronology. Y.Y. carried out age synchronization and firn densification modeling. Y.Y. wrote the manuscript with inputs from all authors.

**Competing interests**

The authors declare that they have no conflict of interest.

**Acknowledgements**

Y.Y. acknowledges funding from Pan Family Postdoctoral Fellowship at Rice University. This work is supported by National Science Foundation Grants ANT-1443306 (University of Maine), ANT-1443276 (Oregon State University), and ANT-1443263 (Princeton University). We thank M. Kalk, J. Edwards, J.E. Lee, L.M. Chimiak, and D.S. Introne for their laboratory assistance. Discussion with J.A. Menking on firn densification modeling improves the manuscript.

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
