# Peer review of "Enhanced Moisture Delivery into Victoria Land, East Antarctica During the Early Last Interglacial: Implications for West Antarctic Ice Sheet Stability"

_Climate of the Past, 2021_

## Referee Comment (RC2)

Yan et al. present atmospheric gas measurements from a blue ice core in the Allan Hills which spans the time period of approximately 110 ka to 250 ka. The atmospheric measurements are high resolution and high quality, and allow a reliable timescale for the gas phase to be developed. The ice-age timescale was previously developed by Spaulding et al., 2013. The primary interpretation in the paper is that there is an order of magnitude change in the gas-age/ice-age difference (dage) from termination II to the last interglacial. The dage drops from a few thousand years to a few hundred, which the authors interpret as being due to a large increase in snowfall in the region during the last interglacial, indicating a significant change in atmospheric circulation during this period of higher global sea level and possibly smaller Antarctic ice sheet volume.

A conclusion of a climatically different Last Interglacial would be interesting and important; however, the authors do not critically evaluate the ice timescale that was previously developed by Spaulding et al. (2013). It is worth noting that the low dage values at ~128 ka are not unique in this core. The authors point out the unphysical negative dage values at ~200 ka; however, the authors don't address that there is a second time period of near-zero dage at ~145 ka (see figure below) in additional to the one at ~128 ka. This is visible in Figure 6 where the ice (red) and gas (black) timescales nearly touch, and it is even more clear plotting up the supplemental data. This is just older than 115-140 ka time period that the authors zoom in on. It seems particularly relevant to the interpretation because an increase in accumulation in both the last interglacial and the previous glacial maximum would have different climatic implications than if it occurred during just the previous interglacial.

The authors accept that the Spaulding et al. tie points with the water isotope isotopes are accurate and precise, yet the Spaulding et al. tie points were not scrutinized for centennial-scale precision because that was not the purpose of that paper. Instead, Spaulding et al. sought to demonstrate an approximately continuous climate record from the core (S27), which they did. Water isotopes records are quite useful for matching broad climate events, but are rarely relied upon for precise synchronization. One of the features which varies substantially among cores is the peak of the Last Interglacial, which is of particular importance to this study. Visual inspection of the S27 and EDC (Figures 1 and 9) and Talos (Spaulding et al., 2013, Figure 7) show broad agreement, but also significant differences on centennial-to-millennial timescales - particularly at the peak Last Interglacial warmth. The authors write "L284-285 unambiguous feature matching (e.g. the distinct MIS5e peak around 128.2 ka)" was not unambiguous to me when I downloaded the S27 dD data (props to Spaulding et al. making it publicly available back in 2013). Trying to make this pick in depth-space, I had to accept a slow rise to the maximum, relative to the EDC dD on depth. This in turn causes sharp changes in the average annual layer thickness. Yes, this could be the result of a change in accumulation, but it can also be an indicator of incorrect tie points. My point here is not that the authors have the tie points wrong, but that to support their conclusion, they need to revisit the ice timescale and demonstrate that the tie points are accurate to a century. There needs to be an uncertainty on the water isotope tie points and the method for developing it described.

Regarding the ice timescale, it is surprising that there is no corroboration of the water isotope tie points with tephra or sulfate/ECM. Tephra matches are the gold standard, and pattern matching of sulfate peaks is the common currency for timescale synchronization. Narcisi et al. (2005; EPSL) found a distinct tephra at 1732.5m, now dated to ~131ka, which would seem to be an obvious candidate even if not perfectly timed to evaluate the dage minimum. Fujita et al. (2015, CP) were able to synchronize EDC and

Dome Fuji throughout this time period, so sulfate/ECM matches may be possible. The S27 average annual layer thickness is only about ½ that of EDC, with only a slightly lower inferred accumulation rate, so the potential for sulfate matching should at least be discussed – there may be something limiting it, but if so, it would be good to discuss what that is and whether that has its own implications for the integrity of the timescale.

The authors also need to discuss why this would indicate a regional signal instead of a local signal. Blue ice areas are particularly sensitive to small changes in climate. One can imagine a slight change in the wind and accumulation pattern could lead to an infinite change in accumulation – from net ablation to net accumulation. Therefore, the paper needs an extended description of the modern accumulation/ablation gradients in the source region. The reference of Kehrl et al. 2018 is provided for the source region, but it should be noted that no radar data have been collected from the actual source region, with the flowline needing to be extended ~10km to the local ice divide. It's also not clear that the position of this ice divide has been steady through time. Taylor Dome is a nearby ice divide for which the glaciological setting was studied extensively to support the ice core drilled there. The accumulation gradient across the dome spans an order of magnitude, 1cm/yr to 10cm/yr (Morse et al., 1999, Geografiska Annaler). And there are extreme km-scale gradients in accumulation rate (potentially 4cm/yr to 0.4cm/yr in 1 to 2 km; Morse et al., 1999, Figure 5). Advection through a gradient like this could produce the type of feature from 126-130 ka, with ice velocities of <1 m/yr. Also interestingly, the pattern of accumulation, based on ice penetrating radar, appears to have reversed between the Holocene and LGM (Morse et al., 1998, GRL).

In summary, the record from the blue ice core S27 is intriguing, but the main conclusion of an accumulation increase at the peak of the Last Interglacial is not sufficiently supported. Further development of the ice timescale is needed to ensure the dage determination is accurate.

---

## Author Comment (AC1)

We thank the three Referees for their constructive comments. We appreciate the generous offer by Dr. Parrenin to provide uncertainty estimates for Δage from IceChrono. We have found the IceChrono code on GitHub and will incorporate the AICC2012 Δage uncertainty in the revised manuscript.

Since all three Referees have commented on our treatment of ice age uncertainty, and Referee #2 and #3 in addition have explicitly raised concerns about the interpretation of the accumulation rate spike around 128 ka, we will organize our response in the following way. We will begin with specifically addressing the uncertainty associated with ice age. Next, we will discuss the possibility of alternative interpretations of the increased accumulation rates. Finally, we will respond to the remainder of comments by individual Referees that do not fit into the first two categories.

**1. Treatment of the ice age uncertainty**

We acknowledge that the ice age uncertainty needs to be considered because the MIS 5e isotope peak in EDC and S27 may not be perfectly synchronous. This offset may become significant when the relative gas age uncertainty becomes sufficiently small, such as during the Δage minimum around 128 ka. We thus proceed to consider the following two sources of ice age uncertainties.

First, we review the synchroneity of temperature variations across Antarctica. Of special importance is the comparison between the $\delta^{18}O_{ice}$ record of Taylor Dome (closest deep core to S27) and that of EDC (the matching target of S27 $\delta D_{ice}$). The assumed synchroneity between S27 and Taylor Dome is supported by their physical proximity (115 km)in the discussion below. Modeling results show that in the event of a collapsed WAIS, both the EDC and Taylor Dome sites are going to experience the same trend in temperature changes (Steig *et al.*, 2015). In addition, the overall deglacial warming is almost synchronous during Termination I in the Taylor Dome and EDC stable water isotope records (Stenni *et al.*, 2011). Both records have an apparent mismatch in peak $\delta^{18}O_{ice}$ around 14 ka, right before the Antarctic Cold Reversal. This offset is about 200 years, translating to the uncertainty of ±100 years associated with the aligning EDC and Taylor Dome ice cores, and by inference, between EDC and S27. Beyond 15 ka, the resolution of Taylor Dome isotope record becomes too low to permit an effective comparison.

Second, we ask how precisely peaks in two time series can be identified and tied. Because we are explicitly targeting the maximum or minimum isotope peaks, the linkage of the observed peaks should be very clear and unambiguous. However, we realize that the peaks in the record were based on discrete sample analysis. In other words, the real peak in the record might not be sampled and captured in the observed peak. Intuitively, the higher the sampling resolution, the smaller the chance of missing the real peak. In the worst-case scenario, the real peak could be located infinitely close to the two samples next to the observed peak. If the sampling resolution is 100 years, for example, then the maximum error associated with identifying the peak in this record is 200 years. In the case of EDC and S27, the average sampling resolution of stable water isotopes during MIS 5e is ~40 and ~20 years, respectively. In attempting to tie the peaks, their respective errors should be added up. In the case of EDC and S27, therefore, an uncertainty of ±60 years related to the identification and matching peaks in different isotope records will be introduced in the revised manuscript.

Taking the two forms of errors in ice timescale into account, we will include an ice age uncertainty of ±160 years in the revised manuscript and update the Δage uncertainty accordingly.

Finally, we wish to take this opportunity to acknowledge that a more refined ice chronology, perhaps made available by absolutely dated tephra layers and synchronizing ion content such as sulfate, as Referee #2 has pointed out, will further improve the manuscript. We hope that this manuscript will stimulate future work on this problem.

**2. Interpretation of the MIS 5e accumulation rate spike**

Referee #2 suggests that the abrupt change in accumulation pattern could be a local effect rather than a broader climate signal, possibly linked to the migration of ice domes and the subsequent changes in accumulation gradient, as some pioneering studies in this region have revealed (Morse *et al.*, 1998; Morse *et al.*, 1999). We are also aware of a recent study by Menking *et al.* (2019) on a horizontal blue ice record drilled from Taylor Glacier (TG). Menking *et al.* calculate the accumulation rate of the TG blue ice record and compare that to the Taylor Dome accumulation rate. They confirm "a spatial gradient in snow accumulation" across the Taylor Dome region. More importantly, their data reveal a reversal in that gradient in LGM compared to MIS 4 [Figure 6 in Menking *et al.* (2019)]. We will acknowledge such possibilities in the revised manuscript.

However, we note that Steig *et al.* (2000) also finds a spike in accumulation rate in Taylor Dome during MIS 5e (Figure 1). Although the timing is not well-constrained, the peak accumulation rate at Taylor Dome is close to 0.08 m·yr$^{-1}$, in good agreement with our estimates of peak accumulation rate at S27. Since the no accumulation estimates is available for TG blue ice record extending back to MIS 5e, we can only compare Taylor Dome and S27 here. We thus consider the increase in accumulation rate during MIS 5e to reflect a regional climatic shift.

[Figure]

Figure 1. Taylor Dome and Vostok accumulation rate reconstruction [Figure 7 in Steig *et al.* (2000)]

Referee #3 further suggests that additional data such as deuterium excess (*d*-excess) could be utilized to test our hypotheses, which we agree and will acknowledge it in our revised manuscript. Indeed, water-tagging experiment in an isotope-enabled model shows that the *d*-excess of the precipitation over the Allan Hills region is most dominated by the moisture source on interannual timescales (Figure 2; Jun Hu, personal communications). Moisture originating from higher latitude has lower *d*-excess values, meaning that all other things being equal, the open-water conditions at the peak of MIS 5e would lead to lower *d*-excess in the S27 record. However, the stable water isotope composition ($\delta D_{ice}$) of the S27 ice core was measured using a mass spectrometer after Cr-pyrolysis at 1050 °C, so no ice core $\delta^{18}O_{ice}$ data is available. We will add these considerations and limitations to our revised manuscript and hope future work can be done to examine the hypothesis put forward in our current study.

Figure 2. Climatological d-excess of water vapor simulated in iCESM (Jun Hu, personal communications).

Now we proceed to address the individual points raised by each Referee that are not related to ice age uncertainty or alternative explanations for the accumulation rate increase.

**3. Response to Referee #1**

We have addressed Dr. Parrenin's main comment about the error estimates above. Below are our responses to the minor comments.

**- l. 25 : "the peak in S27..."**

Thanks for catching that.

**- l. 428-429 : Are you sure 3 ka is enough to re-form the WAIS and/or Ross ice shelf? This could be discussed.**

This is a very good point. It is unlikely that the WAIS could have re-advanced within 3,000 years. Based on the modern observation that the Ross Ice Shelf is fed by both West Antarctic and East Antarctic ice streams (Rignot *et al.*, 2011), it is plausible that the ice shelf recovery originated from East Antarctica. We will add a sentence in the final paragraph of the revised manuscript discussing the recovery of RIS.

**4. Response to Referee #2**

We appreciate the time and efforts by Referee #2 to delve into our data and to raise three very important points. First, a near-zero Δage is present around 145 ka and would imply very large accumulation rate in the glacial period. If this feature is robust, the attribution of elevated accumulation rate during Termination II to the RIS retreat would be weakened. Second, the ice age scale has no error associated with it or independently established age controls (e.g. tephra and sulfate). Third, the accumulation rate change may be a local phenomenon, perhaps related to the migration of accumulation areas. Among them, point #2 and #3 have been addressed in our response above. We therefore discuss the feature of a very small Δage at ~145 ka here.

We underscore the fact that the very small Δage around 145 ka is defined by two $\delta^{18}O_{atm}$-derived, GHG-corrected gas age point at the depth of 136.20 m (140.916 ka) and 139.66 m (143.477 ka). There are only four gas age points between the interval of 140 and 145 ka. In addition, the ice age scale in this interval is constrained by only two tie points, one at 128.32 m (135.808 ka) and the other at 158.69 m (157.096 ka). This is in direct contrast to the small Δage around 128 ka, where 15 $\delta^{18}O_{atm}$ samples are covering the 5,000-year interval from 128 to 133 ka and four $\delta D_{ice}$ tie points lie within the interval between 125 and 130 ka.

To sum up, given the lower temporal resolution of $\delta^{18}O_{atm}$ samples and the fewer ice age tie points around 145 ka, we cannot confidently conclude this small Δage around 145 ka is a robust feature. A similar case can be made for the dip in Δage around 168 ka, where only three $\delta^{18}O_{atm}$ data points provide constraints. In the revised manuscript, we will incorporate these considerations to the text that is currently located between Line 281 and 283. We will also add a new panel to Figure 2 to demonstrate the tie points for ice age scales and

**5. Response to Referee #3**

We thank Referee #3 for the very detailed comments. Before addressing those individual points, we would like to first respond to Referee #3's comments on the impact of gas loss on ice with and without fractures.

[Figure]

Figure 3. Gas loss as observed in ice with and without fractures. Dashed lines are regression lines.

In Figure 3 above, we divide samples into ice with fractures (w/ fracture) and ice without fractures (w/o fracture) and redo the calculation in Figure S4. This yields a slightly steeper slope for ice with fractures (-0.00715±0.00318; 1σ) than for ice without fractures (-0.00654±0.00216; 1σ). The lack of large difference justifies a unified gas loss correction equation. We will add in the revised manuscript that the presence or absence of fractures does not seem to have an impact on the extent of gas loss.

It is curious as to why fractured ice does not appear to experience gas loss differently. One possible explanation is that the gas loss correction here applies to sample measured in 2018. An important assumption is that all data in 2013 were measured on "gas loss-free" ice. This assumption clearly may not hold true for samples below 150 m, as the presence of fractures likely have already impacted the quality of $\delta^{18}O_{atm}$ data back then. In other words, the 5-year gas loss experienced by both fractured and non-fractured ice appears to be the same, but their original status pertaining to gas loss in 2013 was different.

The more detailed points raised by Referee #3 are marked in bold and addressed below. All the text-related comments are fully acknowledged and will be revised accordingly. Here, we focus on the points that substantively impact the interpretation of the S27 record or the presentation of our conclusion.

**Line 43: Specify that it is for both past and future simulations.**

Thanks for pointing this out. We are aware of a recent equilibrium-state simulation of the future warming that shows a widespread retreat of RIS due to the partial collapse of WAIS (Garbe *et al.*, 2020). We will cite this new development in the revised manuscript.

**Line 72: Rephrase the sentence. I guess the missing peak in $\delta^{18}O_{atm}$ is only because no measurements have been done at these depths.**

**Line 73: It is not clear what the $\delta^{18}O_{atm}$ sampling strategy was. Improve the resolution? Complete missing intervals?**

These two comments are related so we will address them together. Part of the initial motivation of this work is indeed finding the missing peak and understanding the stratigraphic integrity of the record. Realizing what could be achieved with a new gas chronology, we eventually decided to measure additional samples from 27 depths above 150 m to further improve the sampling resolution. We will outline the motivation in the revise manuscript with greater clarity.

**Line 145: This sentence suggests that there are also fractures in the ice above 151 m. Are they numerous? Is there an influence on the $\delta^{18}O_{atm}$?**

Yes, fractures are sporadically present between 90 m and 130 m and all ice become fractured once the depth falls below 150 m. We observed an increasing occurrence of fractures with depth between 130 and 150 m. Why the transition of ice quality happens in this depth interval remains not clear. In any case, a $\delta^{18}O_{atm}$ sample requires 20 to 30 g of ice, corresponding to 4 cm in ice length. This is small enough that we may be able to single out the section with no fractures for $\delta^{18}O_{atm}$ analyses even in the transitional zone (130-150 m). A single $CH_4$ sample on the other hand demands a larger sample size (60-70 g) and therefore means longer ice length (10 cm) sample. It is therefore much harder to get a fracture-free ice for $CH_4$.

**Figure 4: Change "per mil" into "‰". You also compare in the main text the $\delta^{18}O_{atm}$ variations to orbital variations. Maybe add the insolation curve on the figure.**

This is a great suggestion. We will add an insolation curve on a second y-axis.

**Figure 5: Add the tie-points and anchor points used for the chronology on the figure. In the caption, precise that $CH_4$ data are from EDC, the $CO_2$ is a composite record and the timescale is AICC2012.**

We will mark tie-points and anchor points in the revised manuscript.

**Line 271: To conclude this part on the gas chronology I missed a sentence on the total uncertainty associated with this new chronology. How much is it?**

We will add more description about the uncertainty associated with the new gas chronology.

**Lines 321-324: I don't know if we can say that the accumulation rate at S27 is comparable to Vostok and EDC. The trend is similar yes but the absolute value not. And how is the 0.02 m.yr-1 value defined?**

We will state that the accumulation rate at S27 is lower than that at Vostok and EDC during glacial periods. The 0.02 $m \cdot yr^{-1}$ is the arithmetic mean value of the Vostok and EDC accumulation rate between 115 and 140 ka, excluding 125 to 132 ka. We realize this is misleading because line 324 states it is "glacial periods", but the interval between 115 and 125 does not technically belong to a glacial period. To avoid confusion, we will not mention this 0.02 $m \cdot yr^{-1}$ in the revised manuscript and instead focus on the relative relationship between S27 and EDC (as well as Vostok) accumulation rates.

**Lines 338-339: Could you support this hypothesis using model comparison?**

Yes, we will add the modeling work by Krinner *et al.* (2007) to support the claim of increased precipitation due to enhanced moisture transport towards the interior of the continent. We note that this work compares the end of the twentieth to the end of twenty-first centuries, but expect the underlying physical mechanism also applies to past climate. In addition, the pattern of precipitation change revealed by the model is spatially heterogenous: while much of Antarctica experiences a higher precipitation, sections of East Antarctic coast (Northern Victoria Land) and West Antarctica receives less precipitation in a warmer climate. That said, for Southern Victoria Land an increased precipitation is observed in the model.

**Lines 345-350: TALDICE's accumulation rate starts to increase earlier than S27 (and is more similar to Vostok and EDC). As for the magnitude, it is much larger for S27 than for TALDICE. The S27 site is already pretty coastal so I would rather say that the peak in accumulation rate at 128 ka reflects more open-ocean conditions than a transition into a coastal site.**

We agree that the timing of the accumulation rate increase Talos Dome precedes the increase in S27. The reason we suggest S27's transitioning into a coastal site is the comparable magnitude of the peak accumulation rate around 128 ka. We will focus on more open-ocean conditions near S27 instead of vaguely calling it a "coastal site" in the revised draft.

**Figure 9: I would have removed the Greenland temperature record and drawn instead the variation in mean ocean temperature from Shackleton et al. (2020). It could also be good to add an insolation curve to have an orbital context to refer to in the discussion. Change "(g) Relative sea-level vs present day".**

We will replace the Greenland temperature curve with the mean ocean temperature series in Shackleton _et al._ (2020) and add a 65 N summer insolation curve. We still believe that the delayed warming of Greenland is important in understanding the sequence of events during Termination II, so the discussion from Line 386 to 388 will be retained.

**In the supplementary:**

**Figure S4: It is not clear if the data presented here are only for non-fractured ice or for both non-fractured and fractured ice. Please indicate if this is non-fractured ice or differentiate the data with two regression lines for the two zones.**

This is from both fractured and non-fractured ice. We have shown in Figure 3 above that the presence or absence of fractures does not appear to impact the gas loss correction.

**References**

Garbe, J., Albrecht, T., Levermann, A., Donges, J.F. and Winkelmann, R., 2020. The hysteresis of the Antarctic ice sheet. *Nature*, *585*(7826), pp.538-544.

Krinner, G., Magand, O., Simmonds, I., Genthon, C. and Dufresne, J.L., 2007. Simulated Antarctic precipitation and surface mass balance at the end of the twentieth and twenty-first centuries. *Climate Dynamics*, *28*(2-3), pp.215-230.

Menking, J.A., Brook, E.J., Shackleton, S.A., Severinghaus, J.P., Dyonisius, M.N., Petrenko, V., McConnell, J.R., Rhodes, R.H., Bauska, T.K., Baggenstos, D. and Marcott, S., 2019. Spatial pattern of accumulation at Taylor Dome during Marine Isotope Stage 4: stratigraphic constraints from Taylor Glacier. *Climate of the Past*, *15*(4), pp.1537-1556.

Morse, D.L., Waddington, E.D., Marshall, H.P., Neumann, T.A., Steig, E.J., Dibb, J.E., Winebrenner, D.P. and Arthern, R.J., 1999. Accumulation rate measurements at Taylor Dome, East Antarctica: Techniques and strategies for mass balance measurements in polar environments. *Geografiska Annaler: Series A, Physical Geography*, *81*(4), pp.683-694.

Morse, D.L., Waddington, E.D. and Steig, E.J., 1998. Ice age storm trajectories inferred from radar stratigraphy at Taylor Dome, Antarctica. *Geophysical Research Letters*, *25*(17), pp.3383-3386.

Rignot, E., Mouginot, J. and Scheuchl, B., 2011. Ice flow of the Antarctic ice sheet. *Science*, *333*(6048), pp.1427-1430.

Shackleton, S., Baggenstos, D., Menking, J.A., Dyonisius, M.N., Bereiter, B., Bauska, T.K., Rhodes, R.H., Brook, E.J., Petrenko, V.V., McConnell, J.R. and Kellerhals, T., 2020. Global ocean heat content in the Last Interglacial. *Nature Geoscience*, *13*(1), pp.77-81.

Steig, E.J., Huybers, K., Singh, H.A., Steiger, N.J., Ding, Q., Frierson, D.M., Popp, T. and White, J.W., 2015. Influence of West Antarctic ice sheet collapse on Antarctic surface climate. *Geophysical Research Letters*, *42*(12), pp.4862-4868.

Steig, E.J., Morse, D.L., Waddington, E.D., Stuiver, M., Grootes, P.M., Mayewski, P.A., Twickler, M.S. and Whitlow, S.I., 2000. Wisconsinan and Holocene climate history from an ice core at Taylor Dome, western Ross Embayment, Antarctica. *Geografiska Annaler: Series A, Physical Geography*, *82*(2-3), pp.213-235.

Stenni, B., Buiron, D., Frezzotti, M., Albani, S., Barbante, C., Bard, E., Barnola, J.M., Baroni, M., Baumgartner, M., Bonazza, M. and Capron, E., 2011. Expression of the bipolar see-saw in Antarctic climate records during the last deglaciation. *Nature Geoscience*, *4*(1), pp.46-49.

---

## Author Response (AR1)

**Author's Response**

We thank the three Referees for their constructive comments. The generous offer by Dr. Parrenin to offer uncertainty estimates for Δage is much appreciated. We have found the IceChrono code on GitHub and have incorporated the AICC2012 Δage uncertainty in the revised manuscript (discussed and cited in the Supplement).

Since all three Referees have commented on our treatment of ice age uncertainty, and Referee #2 and #3 in addition have explicitly raised concerns about the interpretation of the accumulation rate spike around 128 ka, we organize our response in the following way. We begin with specifically addressing the uncertainty associated with ice age. Next, the possibility of alternative interpretations of the increased accumulation rates is discussed. Finally, we respond to the remainder of comments by individual Referees that do not fit into the first two categories.

**1. Treatment of the ice age uncertainty**

We acknowledge that the ice age uncertainty needs to be considered because the MIS 5e isotope peak in EDC and S27 may not be perfectly synchronous. This offset may become significant when the relative gas age uncertainty becomes sufficiently small, such as during the Δage minimum around 128 ka. We thus proceed to consider the following two sources of ice age uncertainties.

First, we review the synchroneity of temperature variations across Antarctica. Of special importance is the comparison between the $\delta^{18}O_{ice}$ record of Taylor Dome (closest deep core to S27) and that of EDC (the matching target of S27 $\delta D_{ice}$). The assumed synchroneity between S27 and Taylor Dome is supported by their physical proximity (115 km) in the discussion below. Modeling results show that in the event of a collapsed WAIS, both the EDC and Taylor Dome sites are going to experience the same trend in temperature changes (Steig *et al.*, 2015). In addition, the overall deglacial warming is almost synchronous during Termination I in the Taylor Dome and EDC stable water isotope records (Stenni *et al.*, 2011). Both records have an apparent mismatch in peak $\delta^{18}O_{ice}$ around 14 ka, right before the Antarctic Cold Reversal. This offset is about 200 years, translating to the uncertainty of ±100 years associated with the aligning EDC and Taylor Dome ice cores, and by inference, between EDC and S27. Beyond 15 ka, the resolution of Taylor Dome isotope record becomes too low to permit an effective comparison.

Second, we ask how precisely peaks in two time series can be identified and tied. Because we are explicitly targeting the maximum or minimum isotope peaks, the linkage of the observed peaks should be very clear and unambiguous. However, we realize that the peaks in the record were based on discrete sample analysis. In other words, the real peak in the record might not be sampled and captured in the observed peak. Intuitively, the higher the sampling resolution, the smaller the chance of missing the real peak. In the worst-case scenario, the real peak could be located infinitely close to the two samples next to the observed peak. If the sampling resolution is 100 years, for example, then the maximum error associated with identifying the peak in this record is 200 years. In the case of EDC and S27, the average sampling resolution of stable water isotopes during MIS 5e is ~40 and ~20 years, respectively. In attempting to tie the peaks, their respective errors should be added up. In the case of EDC and S27, therefore, there is an uncertainty of ±60 years related to the identification and matching peaks in different isotope records.

Taking the two forms of errors in ice timescale into account, we included an ice age uncertainty of ±160 years (2σ) in the revised manuscript and have updated the Δage uncertainty accordingly.

Having now incorporated the uncertainties associated with ice chronology and Δage, we have updated the accumulation rate estimates in the revised manuscript. There is one more modification. In calculating site temperature from $\delta D_{ice}$, we no longer use the $\delta D_{ice}$ value of -257‰ as the calibration point and instead use -270‰. This change is justified in the following ways. First, the $\delta D_{ice}$ observed in S27 during peak LIG is on average about -290‰ (Figure 2 of the manuscript). If we choose -257‰ as the present-day calibration, it implies an unlikely 7.5 °C warming today compared to the LIG. Second, the deuterium excess value of the same surface snow is negative (-5‰). Negative deuterium excess is exceedingly rare in Antarctic snow and in the case of Allan Hills may have been the result of post-depositional processes (Dadic et al, 2015). Third, a depth-gradient of $\delta D_{ice}$ and deuterium excess is observed, reaching -270‰ and 1‰ at the depth of ~0.25 m, respectively. Below this depth, both $\delta D_{ice}$ and deuterium excess show little variability (Figure 2 in Dadic et al, 2015). We interpret these observations to indicate post-depositional alterations to the isotopic compositions of snow above 0.25 m, and therefore use the $\delta D_{ice}$ values below 0.25 m (-270‰) for the isotope-temperature calibration. Nevertheless, using the nominal surface $\delta D_{ice}$ value of -257‰ for temperature calibration systematically increases the accumulation rate estimates by less than 10%. The inference of a large relative increase in accumulation rate at 128 ka is unchanged.

Finally, we wish to take this opportunity to acknowledge that a more refined ice chronology, perhaps made available by absolutely dated tephra layers and synchronizing ion content such as sulfate, as Referee #2 has pointed out, will further improve the manuscript. However, the Allan Hills volcanological records are dominated by regional tephra, with similar composition and contamination by plagioclase crystals from the basement bedrock, complicating the correlation of Allan Hills volcanic record with deep Antarctic ice cores. We hope that this manuscript can stimulate future work on this problem.

**2. Interpretation of the MIS 5e accumulation rate spike**

Referee #2 suggests that the abrupt change in accumulation pattern could be a local effect rather than a broader climate signal, possibly linked to the migration of ice domes and the subsequent changes in accumulation gradient, as some pioneering studies in this region have revealed (Morse *et al.*, 1998; Morse *et al.*, 1999). We are also aware of a recent study by Menking *et al.* (2019) on a horizontal blue ice record drilled from Taylor Glacier (TG). Menking *et al.* calculate the accumulation rate of the TG blue ice record and compare that to the Taylor Dome accumulation rate. They confirm "a spatial gradient in snow accumulation" across the Taylor Dome region. More importantly, their data reveal a reversal in that gradient in LGM compared to MIS 4 [Figure 6 in Menking *et al.* (2019)]. We have acknowledged such possibilities in the revised manuscript.

However, we note that Steig *et al.* (2000) also finds a spike in accumulation rate in Taylor Dome during MIS 5e (Figure 1). Although the timing is not well-constrained, the peak accumulation rate at Taylor Dome is close to 0.08 m·yr$^{-1}$, in good agreement with our estimates of peak accumulation rate at S27. Since the no accumulation estimates is available for TG blue ice record extending back to MIS 5e, we can only compare Taylor Dome and S27 here. We thus consider the increase in accumulation rate during MIS 5e to reflect a

regional climatic shift, but acknowledge in the revised manuscript that the possibility of a local glaciological phenomenon cannot be fully ruled out.

[Figure]

Figure 1. Taylor Dome and Vostok accumulation rate reconstruction [Figure 7 in Steig _et al._ (2000)]

**Climatological water vapor d-excess**

Figure 2. Climatological d-excess of water vapor simulated in iCESM (Jun Hu, personal communications).

Referee #3 further suggests that additional data such as deuterium excess (_d_-excess) could be utilized to test our hypotheses, which we agree and acknowledge it in our revised manuscript. Indeed, water-tagging experiment in an isotope-enabled model shows that the _d_-excess of the precipitation over the Allan Hills region is most dominated by the moisture source on interannual timescales (Figure 2; Jun Hu, personal communications). Moisture originating from higher latitude has lower _d_-excess values, meaning that all other things being equal, the open-water conditions at the peak of MIS 5e would lead to lower _d_-excess in the S27 record. However, the stable water isotope composition ($\delta D_{ice}$) of the S27 ice core was measured using a mass spectrometer after Cr-pyrolysis at 1050 °C, so no ice core $\delta^{18}O_{ice}$ data is

available. We have added these considerations and limitations to our revised manuscript and hope future work can be done to examine the hypothesis put forward in our current study.

Now we proceed to address the individual points raised by each Referee that are not related to ice age uncertainty or alternative explanations for the accumulation rate increase.

**3. Point-by-point response to Referee #1**

We have addressed Dr. Parrenin's main comment about the error estimates above. Below are our responses to the minor comments.

**- l. 25 : "the peak in S27..."**

Corrected.

**- l. 428-429 : Are you sure 3 ka is enough to re-form the WAIS and/or Ross ice shelf? This could be discussed.**

This is a very good point. It is unlikely that the WAIS could have been re-established after collapse within 3,000 years. Based on the modern observation that the Ross Ice Shelf is fed by both West Antarctic and East Antarctic ice streams (Rignot *et al.*, 2011), it is plausible that the ice shelf recovery originated from East Antarctica. We have added a few lines in the discussion section discussing the recovery of RIS.

**4. Response to Referee #2**

We appreciate the time and efforts by Referee #2 to delve into our data and to raise three very important points. First, a near-zero Δage is present around 145 ka and would imply very large accumulation rate in the glacial period. If this feature is robust, the attribution of elevated accumulation rate during Termination II to the RIS retreat would be weakened. Second, the ice age scale has no error associated with it or independently established age controls (e.g. tephra and sulfate). Third, the accumulation rate change may be a local phenomenon, perhaps related to the migration of accumulation areas. Among them, point #2 and #3 have been addressed in our response above. We therefore discuss the feature of a very small Δage at ~145 ka here.

We underscore the fact that the very small Δage around 145 ka is defined by two $\delta^{18}O_{atm}$-derived, GHG-corrected gas age point at the depth of 136.20 m (140.916 ka) and 139.66 m (143.477 ka). There are only four gas age points between the interval of 140 and 145 ka. In addition, the ice age scale in this interval is constrained by only two tie points, one at 128.32 m (135.808 ka) and the other at 158.69 m (157.096 ka). This is in direct contrast to the small Δage around 128 ka, where 15 $\delta^{18}O_{atm}$ samples are covering the 5,000-year interval from 128 to 133 ka and four $\delta D_{ice}$ tie points lie within the interval between 125 and 130 ka.

To sum up, given the lower temporal resolution of $\delta^{18}O_{atm}$ samples and the fewer ice age tie points around 145 ka, we cannot confidently conclude this small Δage around 145 ka is a robust feature. A similar case can be made for the dip in Δage around 168 ka, where only three $\delta^{18}O_{atm}$ data points provide constraints. In the revised manuscript, we have incorporated these considerations to the revised manuscript and marked added in Figure 2 the tie points for ice age scales between S27 and EDC.

**5. Point-by-piont response to Referee #3**

We thank Referee #3 for the very detailed comments. Before addressing those individual points, we would like to first respond to Referee #3's comments on the impact of gas loss on ice with and without fractures.

[Figure]

Figure 3. Gas loss as observed in ice with and without fractures. Dashed lines are regression lines.

In Figure 3 above, we divide samples into ice with fractures (w/ fracture) and ice without fractures (w/o fracture) and redo the calculation in Figure S4. This yields a slightly steeper slope for ice with fractures (-0.00715±0.00318; 1σ) than for ice without fractures (-0.00654±0.00216; 1σ). The lack of large difference justifies a unified gas loss correction equation. Figure S4 in the Supplement has been modified to reflect this concern. We have also stated in the revised manuscript that the presence or absence of fractures does not seem to have an impact on the extent of gas loss.

It is curious as to why fractured ice does not appear to experience gas loss differently. One possible explanation is that the gas loss correction here applies to sample measured in 2018. An important assumption is that all data in 2013 were measured on "gas loss-free" ice. This assumption clearly may not hold true for samples below 150 m, as the presence of fractures likely have already impacted the quality of $\delta^{18}O_{atm}$ data back then. In other words, the 5-year gas loss experienced by both fractured and non-fractured ice appears to be the same, but in 2013, their original status pertaining to gas loss was different.

The more detailed points raised by Referee #3 are marked in bold and addressed below.

**Line 19: Write "during the LIG maximum".**

Done.

**Line 25: Remove the s in "insS27".**

This is a typo and has been corrected.

**Line 43: Specify that it is for both past and future simulations.**

Thanks for pointing this out. We are aware of a recent equilibrium-state simulation of the future warming that shows a widespread retreat of RIS due to the partial collapse of WAIS (Garbe *et al.*, 2020). We have cited this new development in the revised manuscript.

**Figure 1: Add WAIS and Taylor Dome to the map.**

Added with an updated base map.

**Line 57: Remove "that of".**

Done.

**Line 59: Detail the time period covered by the new record.**

Added.

**Line 71: Figure S1 instead of S2.**

In the revised manuscript we have called Figure S1 together with Figure 1 in the Introduction. Now the sequence is correct.

**Line 72: Rephrase the sentence. I guess the missing peak in $\delta^{18}O_{atm}$ is only because no measurements have been done at these depths.**

**Line 73: It is not clear what the $\delta^{18}O_{atm}$ sampling strategy was. Improve the resolution? Complete missing intervals?**

These two comments are related so we address them together. Part of the initial motivation of this work is indeed finding the missing peak and understanding the stratigraphic integrity of the record. Realizing what could be achieved with a new gas chronology, we eventually decided to measure additional samples from 27 depths above 150 m to further improve the sampling resolution. The motivation is better outlined in the revise manuscript.

**Line 75: Specify that new $CO_2$ and $CH_4$ measurements are used to improve the gas chronology between 105-147 ka.**

Added.

**Line 79: Change to "circulation changes and ice shelf / ice-sheet stability during the LIG".**

Done.

**Line 88: Figure S2 instead of S1.**

Same as Line 71; the sequence is now corrected.

**Line 102: Add to the end "to prevent contamination from exchange with ambient air".**

Added.

**Lines 105 and 141: Give the temperature in °C (for consistency).**

The manuscript is now consistently using °C.

**Lines 110-113: Use "$\delta^{18}O$ of $O_2$".**

Done.

**Line 113: Give the equation for gravitational fractionation correction.**

Equations added.

**Line 114: Remove "paleo".**

Removed.

**Line 145: This sentence suggests that there are also fractures in the ice above 151 m. Are they numerous? Is there an influence on the $\delta^{18}O_{atm}$?**

Fractures are sporadically present between 90 m and 130 m and all ice become fractured once the depth falls below 150 m. We observed an increasing occurrence of fractures with depth between 130 and 150 m. Why the transition of ice quality happens in this depth interval remains not clear. In any case, a $\delta^{18}O_{atm}$ sample requires 20 to 30 g of ice, corresponding to 4 cm in ice length. This is small enough that we may be able to single out the section with no fractures for $\delta^{18}O_{atm}$ analyses even in the transitional zone (130-150 m). A single $CH_4$ sample on the other hand demands a larger sample size (60-70 g) and therefore means longer ice length (10 cm) sample. It is therefore much harder to get a fracture-free ice for $CH_4$.

**Line 150: Specify between 115-255 ka.**

Done.

**Line 198: Remove the extra parenthesis for $\delta D_{ice}$.**

Removed.

**Line 205: Wrong units, kg.m$^{-3}$.**

Corrected.

**Line 228: Give the value for $h_{diff}$.**

We have updated the description of how $h_{diff}$ is calculated.

**Line 242: Correct "samples".**

Text corrected.

**Figure 4: Change "per mil" into "‰". You also compare in the main text the $\delta^{18}O_{atm}$ variations to orbital variations. Maybe add the insolation curve on the figure.**

This is a great suggestion. We have added the insolation curve.

**Figure 5: Add the tie-points and anchor points used for the chronology on the figure. In the caption, precise that CH$_4$ data are from EDC, the CO$_2$ is a composite record and the timescale is AICC2012.**

We have marked tie-points and anchor points in the revised manuscript.

**Line 260: Remove "at this peak differ from by 2 ppm".**

Removed.

**Line 271: To conclude this part on the gas chronology I missed a sentence on the total uncertainty associated with this new chronology. How much is it?**

We have added more quantitative description about the uncertainty associated with the new gas chronology.

**Line 281: "Figure 2" not usefull here.**

Removed.

**Lines 321-324: I don't know if we can say that the accumulation rate at S27 is comparable to Vostok and EDC. The trend is similar yes but the absolute value not. And how is the 0.02 m.yr-1 value defined?**

The 0.02 m·yr$^{-1}$ is the arithmetic mean value of the Vostok and EDC accumulation rate between 115 and 140 ka, excluding 125 to 132 ka. We realize this is misleading because line 324 states it is "glacial periods", but the interval between 115 and 125 does not technically belong to a glacial period. To avoid confusion, we do not mention this 0.02 m·yr$^{-1}$ in the revised manuscript and instead focus on the relative relationship between S27 and EDC (as well as Vostok) accumulation rates.

**Lines 338-339: Could you support this hypothesis using model comparison?**

Yes, we have cited the modeling work by Krinner *et al.* (2007) to support the claim of increased precipitation due to enhanced moisture transport towards the interior of the continent. We note that this work compares the end of the twentieth to the end of twenty-first centuries, but expect the underlying physical mechanism also applies to past climate. In addition, the pattern of precipitation change revealed by the model is spatially heterogenous: while much of Antarctica experiences a higher precipitation, sections of East Antarctic coast (Northern Victoria Land) and West Antarctica receives less precipitation in a warmer climate. That said, for Southern Victoria Land an increased precipitation is observed in the model.

**Line 341: Remove the values of the accumulation rate, not usefull.**

Removed.

**Line 344: Delete "apparently".**

Deleted.

**Lines 345-350: TALDICE's accumulation rate starts to increase earlier than S27 (and is more similar to Vostok and EDC). As for the magnitude, it is much larger for S27 than for TALDICE. The S27 site is already pretty coastal so I would rather say that the peak in accumulation rate at 128 ka reflects more open-ocean conditions than a transition into a coastal site.**

We agree that the timing of the accumulation rate increase Talos Dome precedes the increase in S27. The reason we suggest S27's transitioning into a coastal site is the comparable magnitude of the peak accumulation rate around 128 ka. We focus on more open-ocean conditions near S27 instead of vaguely calling it a "coastal site" in the revised draft.

**Figure 9: I would have removed the Greenland temperature record and drawn instead the variation in mean ocean temperature from Shackleton et al. (2020). It could also be good to add an insolation curve to have an orbital context to refer to in the discussion. Change "(g) Relative sea-level vs present day".**

We have replaced the Greenland temperature curve with the mean ocean temperature series in Shackleton *et al.* (2020) and added a 65 N summer insolation curve. We still believe that the delayed warming of Greenland is important in understanding the sequence of events during Termination II, so the discussion in the text is retained.

**Line 414: Remove "(~80 km)".**

Done.

**Line 422: Correct "with a high sea-level stand".**

Corrected.

**In the supplementary:**

**Line 16: Remove "age".**

Deleted.

**Exchange the Figure S1 and the Figure S2 to match the order of citation in the main text.**

Figure S1 is called before Figure S2. The citation sequence is now correct.

**Figure S2: Remove "paleo"**

Removed.

**Figure S3: The $\delta O_2/N_2$ equation for depth > 148 m has to be corrected in $\delta O_2/N_2 = -0.205 \cdot depth(m) + 24.26$. In the caption, give the exact number 0.0067 for the slope.**

Both updated.

**Figure S4: It is not clear if the data presented here are only for non-fractured ice or for both non-fractured and fractured ice. Please indicate if this is non-fractured ice or differentiate the data with two regression lines for the two zones.**

This is from both fractured and non-fractured ice. We have shown in Figure 3 above that the presence or absence of fractures does not appear to impact the gas loss correction.

**Figure S6: We don't see the peak in the C.I. modelled δ$^{15}$N at 128 ka. Adjust the y-axis.**

The y-axis has been adjusted so the full peak at 128 ka can be viewed.

**In Figures S2, S3, S4, S6, change the "per mil" to "‰" for consistency with the main text.**

Done.

**References**

Garbe, J., Albrecht, T., Levermann, A., Donges, J.F. and Winkelmann, R., 2020. The hysteresis of the Antarctic ice sheet. *Nature*, *585*(7826), pp.538-544.

Krinner, G., Magand, O., Simmonds, I., Genthon, C. and Dufresne, J.L., 2007. Simulated Antarctic precipitation and surface mass balance at the end of the twentieth and twenty-first centuries. *Climate Dynamics*, *28*(2-3), pp.215-230.

Menking, J.A., Brook, E.J., Shackleton, S.A., Severinghaus, J.P., Dyonisius, M.N., Petrenko, V., McConnell, J.R., Rhodes, R.H., Bauska, T.K., Baggenstos, D. and Marcott, S., 2019. Spatial pattern of accumulation at Taylor Dome during Marine Isotope Stage 4: stratigraphic constraints from Taylor Glacier. *Climate of the Past*, *15*(4), pp.1537-1556.

Morse, D.L., Waddington, E.D., Marshall, H.P., Neumann, T.A., Steig, E.J., Dibb, J.E., Winebrenner, D.P. and Arthern, R.J., 1999. Accumulation rate measurements at Taylor Dome, East Antarctica: Techniques and strategies for mass balance measurements in polar environments. *Geografiska Annaler: Series A, Physical Geography*, *81*(4), pp.683-694.

Morse, D.L., Waddington, E.D. and Steig, E.J., 1998. Ice age storm trajectories inferred from radar stratigraphy at Taylor Dome, Antarctica. *Geophysical Research Letters*, *25*(17), pp.3383-3386.

Rignot, E., Mouginot, J. and Scheuchl, B., 2011. Ice flow of the Antarctic ice sheet. *Science*, *333*(6048), pp.1427-1430.

Shackleton, S., Baggenstos, D., Menking, J.A., Dyonisius, M.N., Bereiter, B., Bauska, T.K., Rhodes, R.H., Brook, E.J., Petrenko, V.V., McConnell, J.R. and Kellerhals, T., 2020. Global ocean heat content in the Last Interglacial. *Nature Geoscience*, *13*(1), pp.77-81.

Steig, E.J., Huybers, K., Singh, H.A., Steiger, N.J., Ding, Q., Frierson, D.M., Popp, T. and White, J.W., 2015. Influence of West Antarctic ice sheet collapse on Antarctic surface climate. *Geophysical Research Letters*, *42*(12), pp.4862-4868.

Steig, E.J., Morse, D.L., Waddington, E.D., Stuiver, M., Grootes, P.M., Mayewski, P.A., Twickler, M.S. and Whitlow, S.I., 2000. Wisconsinan and Holocene climate history from an ice core at Taylor Dome, western Ross Embayment, Antarctica. *Geografiska Annaler: Series A, Physical Geography*, *82*(2-3), pp.213-235.

Stenni, B., Buiron, D., Frezzotti, M., Albani, S., Barbante, C., Bard, E., Barnola, J.M., Baroni, M., Baumgartner, M., Bonazza, M. and Capron, E., 2011. Expression of the bipolar see-saw in Antarctic climate records during the last deglaciation. *Nature Geoscience*, *4*(1), pp.46-49.

---

## Referee Report (RR1)

Review of Yan et al. revisions

Yan et al. have provided a revised manuscript "Enhanced Moisture Delivery into Victoria Land, East Antarctica During the Early Last Interglacial: Implications for West Antarctic Ice Sheet Stability". The revisions include improved figures and fuller descriptions of the limitations of their conclusions. However, I think the issue of the ice timescale tie points requires a fuller analysis in the main text. In particular, the inference of accumulation rate is highly dependent on the tie points between 128 and 129 ka. These are based on a visual match of a noisy isotope peak in S27 to a smooth peak EDC. The S27 isotope record has 3 different peaks that could plausibly be considered the maximum after applying a smoothing 5 data points.

To illustrate the importance of these tie points, the figure below shows the change in delta-age that occurs if you remove the 3 tie points between 128 and 129 ka in the bottom panel (and use linear interpolation between the tie points). The small delta-age which gives rise to the inference of the high accumulation no longer exists. The removal of the tie points also affects the look of the isotope record, which is plotted in the top panel. Can the resulting isotope record be excluded from consideration? If so, on what basis? I want to emphasize that this is not the only plausible shift in the ice timescale that would impact the delta-age; it is just one that was relatively simple to do as a reviewer. It strikes me that the delta-age uncertainties in Figure 7 are missing an important source of uncertainty and are considerably larger than shown.

I think the paper would be much improved if there was a quantitative way of aligning (and assessing the alignment of) the S27 isotope record with the EDC isotope record. Lee et al. (2020 in Climate of the Past) use a matching method for the methane record of Roosevelt Island which could be applicable here. The inference of high accumulation during the Last Interglacial has the potential to be very impactful which is why I think the conclusion warrants substantial scrutiny.

---

## Author Response (AR2)

**Author's Response**

Dear Dr. Wolff,

We thank Referee #2 for the effort to re-review the manuscript and appreciate your suggestions on the ice chronology. Per your recommendation, we have added a paragraph along with two figures in the Supplement (Line 71-82) to discuss the potential impact of mismatched $\delta D_{ice}$ in the original ice timescale. It is demonstrated that in order for $\Delta$age to remain persistently above ~2,000 years during the LIG, there needs to be a mismatch in $\delta D_{ice}$ peaks by ~1,000 years. While such a mismatch is not likely, we still acknowledge its possibility and make the following admission in the main text:

"… we acknowledge that given the noisy nature of the S27 $\delta D_{ice}$ records (Figure 2 and Figure S7), it is possible that the $\Delta$age—and by inference the ice accumulation rates—could have larger errors than reported here, which we discuss in greater detail in the Supplement."

The two new figures added to the Supplement are also attached to this letter.

Finally, we agree that our work here is not the smoking gun on this important topic. Rather, it is our hope that the hypothesis presented in this work will spark future investigations such as ice core deuterium excess and major ion during the LIG to better understand the behaviors of the West Antarctic Ice Sheet.

Best regards,

Yuzhen Yan

[Figure]

**Figure S7. Evaluating alternative δD$_{ice}$ tie-points around 128 ka and its impact on ice chronology.** The top and middle panel are a zoom-in view of Figure 2 in the main text, with tie-points circled by solid lines. The lower panel shows the same δD$_{ice}$ record under a different tie-point scheme: the δD$_{ice}$ peak at 128.01 ka in the middle panel (dashed square) is tied to the EDC δD$_{ice}$ peak at 128.33 ka. This new tie-point (square) leads to an older ice age at the same depth and hence larger Δage and smaller accumulation rates. In order for Δage to remain unchanged across the MIS 5e (Figure S8), the δD$_{ice}$ peak around 127.25 ka (dashed triangle) needs to be tied to the 128.33 ka EDC δD$_{ice}$ peak.

[Figure]

**Figure S8. Δage estiamtes under different tie-point scenarios.** Solid red: original tie-points adopted by Spaulding et al (2013) and used in this study (same as Figure 7 in the main text). Dashed red: S27 $\delta D_{ice}$ at 128.01 ka tied to the EDC $\delta D_{ice}$ at 128.33 ka (square in Figure S7). Solid blue: S27 $\delta D_{ice}$ at 127.25 ka tied to the EDC $\delta D_{ice}$ at 128.33 ka (triangle in Figure S7).